# MRGPRX4 is a bile acid receptor for human cholestatic itch

Huasheng Yu[1,2,3], Tianjun Zhao[1,2,3], Simin Liu[1], Qinxue Wu[4], Omar Johnson[4], Zhaofa Wu[1,2], Zihao Zhuang[1], Yaocheng Shi[5], Luxin Peng[5], Renxi He[1,2], Yong Yang[6], Jianjun Sun[7], Xiaoqun Wang[8], Haifeng Xu[9], Zheng Zeng[10], Peng Zou[5], Xiaoguang Lei[3,5], Wenqin Luo[4]*, Yulong Li[1,2,3,11]*

[1]State Key Laboratory of Membrane Biology, Peking University School of Life Sciences, Beijing, China; [2]PKU-IDG/McGovern Institute for Brain Research, Beijing, China; [3]Peking-Tsinghua Center for Life Sciences, Beijing, China; [4]Department of Neuroscience, Perelman School of Medicine, University of Pennsylvania, Philadelphia, United States; [5]Department of Chemical Biology, College of Chemistry and Molecular Engineering, Peking University, Beijing, China; [6]Department of Dermatology, Beijing Key Laboratory of Molecular Diagnosis on Dermatoses, Peking University First Hospital, Peking University, Beijing, China; [7]Department of Neurosurgery, Peking University Third Hospital, Peking University, Beijing, China; [8]State Key Laboratory of Brain and Cognitive Science, CAS Center for Excellence in Brain Science and Intelligence Technology (Shanghai), Institute of Biophysics, Chinese Academy of Sciences, Beijing, China; [9]Department of Liver Surgery, Peking Union Medical College Hospital, Chinese Academy of Medical Sciences, Peking Union Medical College, Beijing, China; [10]Department of Infectious Diseases, Peking University First Hospital, Beijing, China; [11]Chinese Institute for Brain Research, Beijing, China

*For correspondence:
luow@pennmedicine.upenn.edu
(WL);
yulongli@pku.edu.cn (YL)

Competing interests: The
authors declare that no
competing interests exist.

Reviewing editor: David D
Ginty, Harvard Medical School,
United States

**Abstract** Patients with liver diseases often suffer from chronic itch, yet the pruritogen(s) and receptor(s) remain largely elusive. Here, we identify bile acids as natural ligands for MRGPRX4. MRGPRX4 is expressed in human dorsal root ganglion (hDRG) neurons and co-expresses with itch receptor HRH1. Bile acids elicited $Ca^{2+}$ responses in cultured hDRG neurons, and bile acids or a MRGPRX4 specific agonist induced itch in human subjects. However, a specific agonist for another bile acid receptor TGR5 failed to induce itch in human subjects and we find that human TGR5 is not expressed in hDRG neurons. Finally, we show positive correlation between cholestatic itch and plasma bile acids level in itchy patients and the elevated bile acids is sufficient to activate MRGPRX4. Taken together, our data strongly suggest that MRGPRX4 is a novel bile acid receptor that likely underlies cholestatic itch in human, providing a promising new drug target for anti-itch therapies.

DOI: https://doi.org/10.7554/eLife.48431.001

## Introduction

Chronic itch, or pruritus, is a severe and potentially debilitating clinical feature associated with many dermatological and systemic conditions (*Koch et al., 2018*), severely affecting quality of life and potentially leading to lassitude, fatigue, and even depression and suicidal tendencies (*Tajiri and Shimizu, 2017*). The most well-characterized itch receptors are the H1 and H4 histamine receptors (HRH1 and HRH4) (*Thurmond et al., 2008*). Although antihistamines, which act by inhibiting histamine receptors, are generally effective at relieving itch symptoms induced by inflammation and

allergens, these compounds are usually ineffective at treating chronic itch caused by systemic diseases and most skin disorders. To date, no effective treatment is available for treating histamine-resistant itch (*Tajiri and Shimizu, 2017*).

A high percentage of patients with systemic liver failure develop itch with cholestatic symptoms (*Beuers et al., 2014*). For example, the prevalence of itch is as high as 69% among patients with primary biliary cirrhosis, and severe itch is an indication for liver transplantation (*Imam et al., 2012*). Moreover, itch occurs in more than half of pregnant woman with intrahepatic cholestasis of pregnancy, a condition that has been associated with an increased risk of preterm delivery, perinatal mortality, and fetal distress (*Jenkins and Boothby, 2002*).

Several medications have been tested for treating cholestatic itch, including ursodeoxycholic acid (UDCA), cholestyramine, and rifampicin; however, these compounds either are ineffective or induce severe side effects (*Imam et al., 2012*). Therefore, safe and effective treatments for cholestatic itch are urgently needed, and identifying the underlying molecular mechanisms—particularly the receptor and ligand—is the essential first step.

Although the link between cholestasis and itch was first described more than 2000 years ago (*Kremer et al., 2011*), the detailed mechanisms underlying cholestatic itch remain unidentified. To date, a handful of molecules have been proposed as the pruritogens that mediate cholestatic itch, including bile acids, bilirubin, lysophosphatidic acid, autotaxin, and endogenous opioids (*Beuers et al., 2014*). With respect to the cognate receptor for the pruritogen, a few receptors have been proposed, albeit based primarily on rodent models. For example, the membrane-bound bile acid receptor TGR5 has been reported to mediate bile acid–induced itch in mice (*Alemi et al., 2013*; *Lieu et al., 2014*). However, a recent study found that administering TGR5-selective agonists failed to elicit an itch response in mouse models of cholestasis (*Cipriani et al., 2015*), and TGR5-specific agonists in recent clinical trials have not been reported to have itch side effect (*Hodge et al., 2013*), raising doubts regarding whether TGR5 is indeed the principal mediator for cholestatic itch. Recently, Meixiong et al. reported that mouse Mrgpra1 and human MRGPRX4 can be activated by bilirubin, a compound that serves as one of the pruritogens in cholestatic itch in mice (*Meixiong et al., 2019a*). Meixiong et al. also reported that MRGPRX4 can be activated by bile acids and showed that humanized MRGPRX4 transgenic mice exhibited itch in response to bile acid injection and in a mouse model of cholestasis (*Meixiong et al., 2019b*). However, evidence in demonstrating the direct role of MRGPRX4 in mediating cholestatic pruritus in human subjects is still weak. In addition, whether bile acid receptor TGR5 participates in cholestatic itch in human also remains unclear.

Through a different strategy, we also identified that MRGPRX4 is the receptor activated by bile acids. Furthermore, we provide comprehensive molecular, cellular, behavioral and clinical evidences that MRGPRX4 is a bile acid receptor for cholestatic itch in human. We specifically focused our search on genes that are selectively expressed in the human dorsal root ganglia (DRG), where the cell bodies of primary itch-sensing neurons are located. Our screening identified that bile acids from bile extract could activate MRGPRX4, one of the G protein coupled receptors (GPCRs) highly expressed in human DRGs. We further showed that MRGPRX4 is expressed selectively in a small subset of human DRG neurons, and bile acids directly trigger intracellular $Ca^{2+}$ increase in human neurons and action potentials in MRGPRX4 transfected rat DRG neurons. In addition, intradermal injection of both bile acids and the MRGPRX4-specific agonist nateglinide induce detectable itch in human subjects, and this bile acid-induced itch is histamine-independent. We also investigated the role of another bile acid receptor, TGR5, in cholestatic itch in human. Surprisingly, application of specific agonist for TGR5, previously implicated in cholestatic itch in mice, failed to elicit $Ca^{2+}$ response in cultured hDRG neurons, nor did it induce pruritus in human subjects. In situ hybridization and immunostaining results revealed that unlike mouse TGR5 (mTGR5) expressing in mouse DRG neurons, hTGR5 is selectively expressed in satellite glial cells, likely accounting for the inter-species difference in function. Finally, we found that plasma bile acid levels are well correlated with itch sensation in cholestatic patients and that this elevated bile acid level is sufficient to activate MRGPRX4. Taken together, our results provide compelling evidence that the ligand-receptor pair of bile acids and MRGPRX4 is likely to be one of the critical mediators for human cholestatic itch.

## Results

### MRGPRX4 is activated by bile extract

DRG neurons are primary somatosensory neurons that express a variety of receptors and ion channels for detecting both extrinsic and intrinsic stimuli (*Belmonte and Viana, 2008*). To identify a receptor in mediating cholestatic itch in human, we reason that this candidate receptor could be expressed in human DRG neurons and activated by bile extracts. Since the majority of itch receptors identified to date belong to the G protein–coupled receptor (GPCR) superfamily (*Dong and Dong, 2018*), we analyzed two published transcriptomics datasets compiled from a variety of human tissues (*Flegel et al., 2013*; *Flegel et al., 2015*), specifically focusing on GPCRs. Among the 332 transcripts that are enriched in the human DRG (*Supplementary file 1*), we identified the following seven highest-enriched GPCRs: GPR149, MRGPRX4, GPR139, GPR83, MRGPRE, MRGPRX1, and MRGPRD (*Liu et al., 2009*; *Liu et al., 2012*) (*Figure 1a* and *Supplementary file 2*). Next, we cloned and expressed these candidate receptors in HEK293T cells (*Figure 1—figure supplement 1a,b*), finding that all seven receptors were expressed at the plasma membrane (*Figure 1—figure supplement 1b*). We measured the activation of each receptor by bovine bile extract using two reporter assays, the Gs-dependent luciferase assay (*Hall et al., 2012*) and the Gq-dependent TGFα shedding assay (*Inoue et al., 2012*) (*Figure 1b, c*). No signal was detected with the Gs-dependent luciferase assay (*Figure 1b*). Interestingly, bile extract elicited a significant increase in reporter activity in cells expressing MRGPRX4 measured using the TGFα shedding assay, but had no effect on cells expressing the other six GPCRs (*Figure 1c*). These results suggest that MRGPRX4 is activated by one or more compounds present in bile extract, and that MRGPRX4 likely signals through the Gq but not the Gs pathway. Further experiments revealed that bovine, porcine, and human bile extract activate MRGPRX4 to a similar extent in a dose-dependent manner (*Figure 1d*); in contrast, extracts obtained from bovine brain, spleen, heart, kidney, and liver tissues induced no detectable signal on MRGPRX4-expressing cells (*Figure 1e*). Taken together, these results suggest that MRGPRX4 is potently activated by bile extract and active compound(s) is/are highly enriched in bile extract.

### Identifying which in bile extract activate MRGPRX4

Next, to identify the component(s) in bovine bile extract that activate(s) MRGPRX4, we separated the extract into six fractions using silica gel column chromatography (*Figure 2a*). Each fraction was then applied to MRGPRX4-expressing HEK293T cells, and MRGPRX4 activation was measured using the TGFα shedding assay. Among the six fractions tested, fraction four caused the strongest activation of MRGPRX4, whereas fractions 1 and 6 caused the weakest activity (*Figure 2b*), indicating that the active component(s) are mainly present in fraction 4. Mass spectrometry of fractions 4 and 6 revealed a peak enriched specifically in fraction 4 (*Figure 2c*); this peak corresponded to ions with an m/z value of 410.3265 in the positive ion mode and was annotated to prostaglandin F2α diethyl amide and/or dihydroxy bile acids. Further experiments using $^1$H-NMR revealed that two pure dihydroxy bile acids—deoxycholic acid (DCA) and chenodeoxycholic acid (CDCA)—produced peaks that were identical to the peaks in fraction 4 (*Figure 2d*). These results suggest that DCA and/or CDCA are enriched in the active fraction of bile extract and may be the key compounds that activate MRGPRX4. In addition to fraction 4, other fractions, including fractions 2, 3 and 5, showed weaker but also significant activity. Using $^1$H-NMR, we found that fractions 2 and 3 contained the characteristic peaks of bile acids. (*Figure 2—figure supplement 1*), suggesting that the activity may also come from the bile acids in these fractions.

### Characterization of bile acids: MRGPRX4 activation and the downstream signaling

To further characterize the efficacy and potency of DCA, CDCA, other bile acids, and their derivatives in activating MRGPRX4, we systematically measured their ability to activate MRGPRX4 in HEK293T cells, using TGFα shedding assay and FLIPR (fluorescent imaging plate reader) Ca$^{2+}$ assay. All of the bile acids tested activated MRGPRX4 to some extent; DCA had the highest potency measured with both assays, with an EC$_{50}$ value of 2.7 μM and 2.6 μM in the TGFα shedding and FLIPR

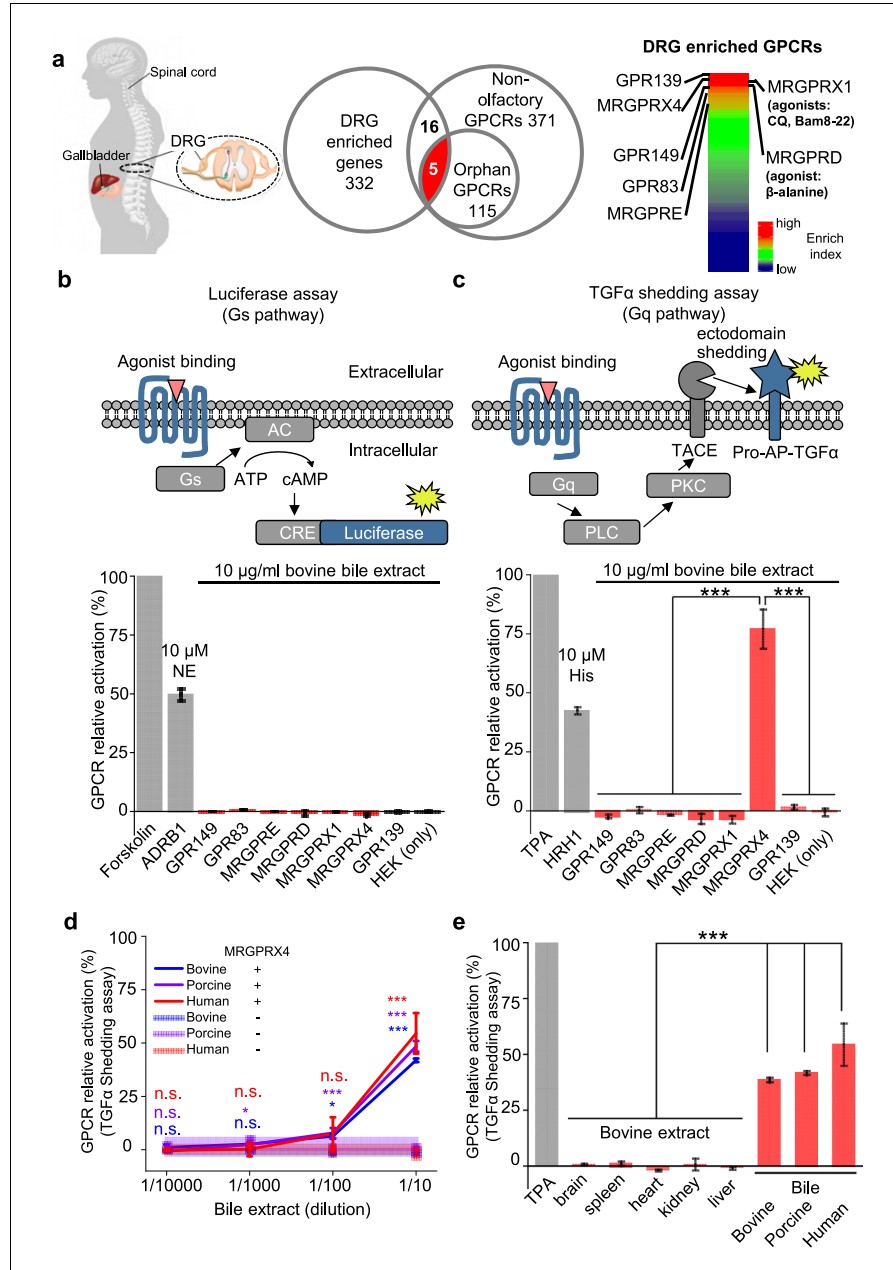

**Figure 1.** MRGPRX4 is activated by bile extract. (**a**) Flow chart for the strategy used to identify GPCRs enriched in human DRG. Transcriptome analysis of DRG and other tissues (trigeminal ganglia, brain, colon, liver, lung, skeletal muscle, and testis) revealed 332 transcripts with high expression in the DRG. The top seven GPCRs are listed. See also *Supplementary files 1* and *2*. Gene expression data were obtained from *Flegel et al. (2013)*. (**b** and **c**) Activation of MRGPRX4 by bovine bile extract. The diagrams at the top depict the reporter gene assays used to measure GPCR activation via Gs-dependent (**b**) and Gq-dependent (**c**) pathways. The seven GPCRs identified in (**a**) were tested, revealing that MRGPRX4-expressing HEK293T cells are activated by bile extract via the Gq-dependent pathway. Forskolin and TPA were used as positive controls for activating Gs- and Gq-dependent signaling, respectively. The responses obtained from the tested GPCRs were normalized to the responses induced by respective positive controls. As positive controls for detecting GPCR activation, separate cells were transfected with ADRB1 and stimulated with 10 μM norepinephrine (NE) (**b**) or transfected with HRH1 and stimulated with 10 μM histamine (His) (**c**). 'HEK (only)' refers to non-transfected cells. n = 3 experiments performed in triplicate. (**d**) Concentration-response curve for the activation of MRGPRX4 by bovine bile extract, porcine bile extract, and human bile measured using the TGFα shedding assay. The bovine and porcine bile extract solutions were diluted 1 :10 from a 100 μg/ml stock solution, and the human bile solution was diluted 1:10 from crude human bile. n = 2

*Figure 1 continued on next page*

*Figure 1 continued*
experiments performed in triplicate. (**e**) MRGPRX4 is activated selectively by bovine, porcine, and human bile extracts, but not by bovine brain, spleen, heart, kidney, or liver tissue extracts. The data for porcine and human bile are reproduced from (**d**). n = 2 experiments performed in triplicate. All error bars represent the s.e.m.. Student's *t*-test, *p<0.05, ***p<0.001, and n.s. not significant (p>0.05).
DOI: https://doi.org/10.7554/eLife.48431.002
The following figure supplement is available for figure 1:

**Figure supplement 1.** Construct design and surface expression of candidate GPCRs in HEK293T cells.
DOI: https://doi.org/10.7554/eLife.48431.003

assays, respectively; cholic acid (CA), CDCA, and lithocholic acid (LCA)—three close analogs of DCA—were less potent (*Figure 3a–c*). Based on the structural differences between DCA and the less potent bile acids, we reasoned that hydroxylation at position of R1 and/or R2, as well as taurine/glycine conjugation at position R3, is important for specific bile acids to activate MRGPRX4 (*Figure 3c*).

Next, we examined the potential signaling events downstream of bile acid-induced MRGPRX4 activation by measuring intracellular $Ca^{2+}$ concentration ($[Ca^{2+}]_i$) in MRGPRX4-expressing HEK293T cells loaded with Fluo-8 AM, a fluorescent $Ca^{2+}$ indicator. We found that DCA, CA, CDCA, and LCA induced a robust fluorescence response in these cells (*Figure 3d–f*), and pretreating the cells with the phospholipase C inhibitor U73122 significantly reduced the DCA-evoked $Ca^{2+}$ signals; in contrast, the Gβγ inhibitor gallein had no effect on DCA-evoked signaling (*Figure 3g,h*). Taken together, these results indicate that a Gq-dependent signaling pathway involving phospholipase C is downstream to MRGPRX4 activation by bile acids.

Interestingly, even though MRGPRX1, MRGPRX2, and MRGPRX3 are close analogs of MRGPRX4, none of these receptors was activated by bile acids, even at 100 µM concentration (*Figure 3—figure supplement 1a–e*). We therefore investigated the putative ligand-binding sites in MRGPRX4 by comparing the primary amino acid sequence of MRGPRX4 with these three analogs (*Figure 3i*). We identified amino acid residues that are conserved in MRGPRX1, MRGPRX2, and MRGPRX3 but not in MRGPRX4 and mutated these residues, once per time, to an alanine residue in MRGPRX4. We found that mutating amino acids 159, 180, and 235 reduced the receptor's affinity for DCA (*Figure 3j*), without diminishing protein trafficking to the cell membrane (*Figure 3k*); thus, these three sites may play a critical role in the binding of bile acids to MRGPRX4. We noticed that the membrane expression level of the P180A mutant is nearly 3-fold higher than the other mutants or wild type, which may be due to a favorable genomic insertion site or multiple insertion copies when we generated the stable cell line via piggyBac transposon system (*Yusa et al., 2011*). In addition, we examined whether mouse and/or rat Mrgpr family members also respond to bile acids. Intriguingly, bile acids failed to activate any mouse or rat Mrgpr members tested (*Figure 3—figure supplement 1g,h*), suggesting that the ability of MRGPRX4 to sense bile acids may be a new functional addition during evolution.

## MRGPRX4 is expressed in a subset of hDRG neurons

Next, we examined endogenous expression pattern of MRGPRX4 in hDRGs. We performed in situ hybridization using a digoxigenin-labeled riboprobe against *MRGPRX4* mRNA, and found that *MRGPRX4* mRNA is expressed in only ~ 6–8% of hDRG neurons (*Figure 4a,c*); similar results were obtained with immunofluorescence using an MRGPRX4-specific antibody (*Figure 4b,c* and *Figure 4—figure supplement 1*). Morphologically, these MRGPRX4-expressing neurons are small-diameter neurons, with a diameter of approximately 50 µm, which is similar to small-diameter neurons that express the neurotrophic tyrosine kinase receptor type 1 (TrkA) (*Figure 4d*), suggesting a function in nociception and/or pruriception (*Patapoutian and Reichardt, 2001*).

To further characterize the molecular profile of these MRGPRX4-positive hDRG neurons, we performed triple-labeling of MRGPRX4 and two additional molecular markers using RNAscope in situ hybridization (*Figure 4e*). Since it is hard to tell a real MRPGRX4+ positive neuron from background at the low magnification due to its weak signal (data not shown), we mostly took high-magnified single-neuron images for quantitative analysis. Our analysis revealed that > 90% of MRGPRX4-positive neurons also express the histamine receptor HRH1, a well-characterized itch receptor in humans

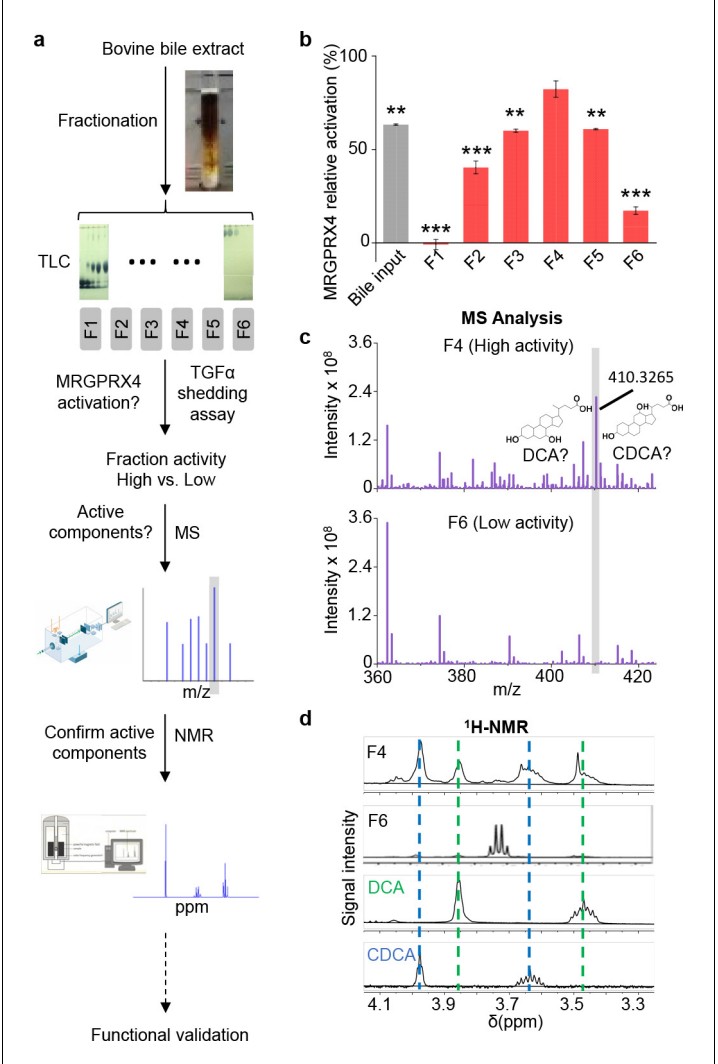

**Figure 2.** Identification of the active components in bile extract that activate MRGPRX4. (**a**) Flow chart depicting the strategy for isolating and identifying candidate MRGPRX4 ligands in bovine bile extract. F1 through F6 indicate the six fractions used in subsequent experiments. (**b**) Activation of MRGPRX4 by bile extract fractions F1 through F6; fraction F4 has the highest activity. The data represent one experiment performed in triplicate. All error bars represent the s.e.m.. Student's *t*-test, **p<0.01. ***p<0.001 versus fraction F4. (**c**) MS analysis of fractions F4 and F6 (which showed high and weak activity, respectively). The selectively enriched peak in fraction F4 at molecular weight 410.3265 corresponds to the bile acids DCA and CDCA. (**d**) $^1$H-NMR analysis of fractions F4 and F6 using purified DCA and CDCA as controls.
DOI: https://doi.org/10.7554/eLife.48431.004

The following figure supplement is available for figure 2:

**Figure supplement 1.** $^1$H-NMR analysis of bile acids in fractions F1, F2, F3, F4, and F6.
DOI: https://doi.org/10.7554/eLife.48431.005

(*Han et al., 2006*), and TRPV1 (transient receptor potential cation channel subfamily V member 1) (*Figure 4f,g*), which functions downstream of Mrgprs and histamine receptors (*Imamachi et al., 2009*; *Wilson et al., 2011*). Interestingly, the majority of MRGPRX4-expressing neurons also co-express Na$_v$1.7 voltage-gated sodium channel, the peptidergic marker CGRP (calcitonin gene-related peptide), and TrkA (*Usoskin et al., 2015*; *Li et al., 2016*) (*Figure 4f,g*). These results suggest that MRGPRX4 is specifically expressed in a subset of small diameter peptidergic hDRG neurons.

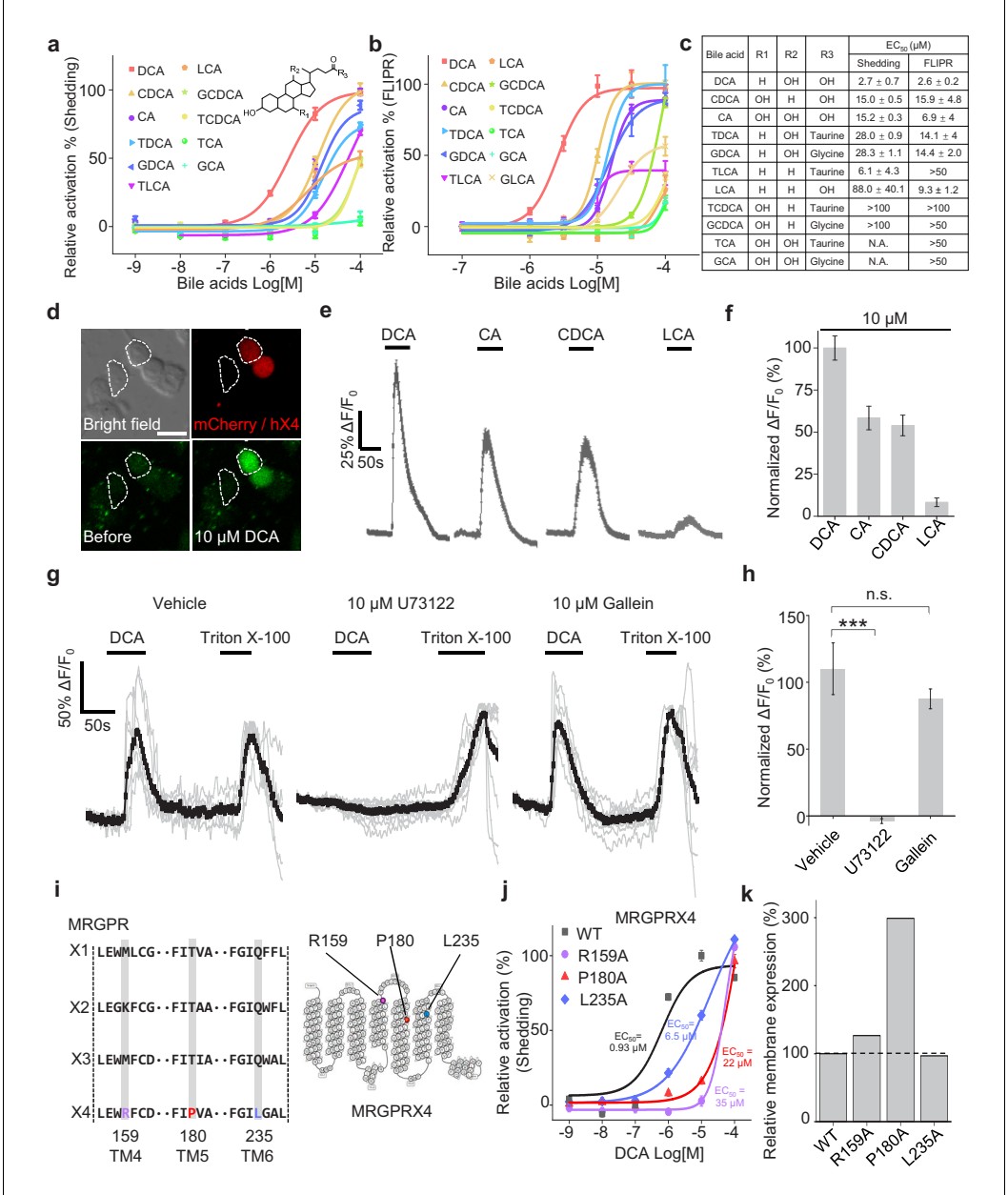

**Figure 3.** Functional characterization and molecular profiling of bile acids as ligands for MRGPRX4. (**a–c**) Dose-dependent activation of MRGPRX4 by various bile acids and their derivatives. MRGPRX4 activation was measured using the TGFα shedding assay (**a**) or the FLIPR assay (**b**) see Materials and methods) in MRGPRX4-expressing HEK293T cells; n = 1 experiment performed in triplicate. The general structure of the bile acids and derivatives is shown in (**a**), and the respective potencies of the bile acids/derivatives are listed in (**c**). (**d–f**) Activation of MRGPRX4 by various bile acids in cells loaded with the Ca²⁺ indicator Fluo-8 AM. (**d**) Representative images of MRGPRX4-expressing HEK293T cells (shown by mCherry fluorescence) before and after application of 10 μM DCA. (**e**) Representative traces of Ca²⁺ responses induced by application of 10 μM DCA, CA, CDCA, or LCA. n = 50 cells each. (**g–h**) MRGPRX4 is coupled to the Gq-PLC-Ca²⁺ signaling pathway. DCA (10 μM) evoked a robust Ca²⁺ signal in MRGPRX4-expressing HEK293T cells (**g**), left); this response was blocked by pretreating cells for 30 min with the PLC inhibitor U73122 (**g**, middle), but not the Gβγ inhibitor gallein (**g**, right). Triton X-100 was used as a positive control. The summary data are shown in (**h**); n = 7–10 cells each. Student's t-test, ***p<0.001, and n.s. not significant (p>0.05). (**i–k**) Identification of key residues in MRGPRX4 that mediate ligand binding and receptor activation. (**i**) Primary sequence alignment of the human MRGPRX1, MRGPRX2, MRGPRX3, and MRGPRX4 proteins. The positions of the three amino acids in MRGPRX4 that were mutated to alanine are shown at the right. (**j**) Dose-dependent activation of wild-type (WT) MRGPRX4 and three MRGPRX4 mutants with the indicated point

*Figure 3 continued on next page*

*Figure 3 continued*

mutations was measured using the TGFα shedding assay. n = 1 experiment performed in triplicate. (**k**) Plasma membrane expression of Myc-tagged WT and mutant MRGPRX4 was measured using an anti-Myc antibody and normalized to WT MRGPRX4 expression. All error bars represent the s.e.m.

DOI: https://doi.org/10.7554/eLife.48431.006

The following figure supplement is available for figure 3:

**Figure supplement 1.** Human MRGPRX4, but not human MRGPRX1-3 or mouse and rat Mrgpr family members, are activated by bile acids.

DOI: https://doi.org/10.7554/eLife.48431.007

## MRGPRX4 mediates bile acid induced activation of DRG neurons

Next, we tested whether MRGPRX4 in DRG neurons can be activated by bile acids. Because bile acids failed to induce a detectable $Ca^{2+}$ signal in cultured rat DRG neurons (*Figure 5—figure supplement 1a–c*), we electrically transfected and expressed the human MRGPRX4 in cultured rat DRG neurons. Bile acids triggered a robust $Ca^{2+}$ response in MRGPRX4-expressing rat DRG neurons but not in non-transfected control cells (*Figure 5a–d*). We also detected bile acid-induced action potentials in MRGPRX4-expressing rat DRG neurons but not in non-transfected control cells by whole cell recordings (*Figure 5e,f*, *Figure 5—figure supplement 1e,f*). These results demonstrate that MRGPRX4 expressed in rat DRG neurons could mediate the bile acid-–induced activation. Consistent with our finding that DCA is a more potent agonist of MRGPRX4 than CA, DCA induced a significantly larger $Ca^{2+}$ response and activated a larger number of MRGPRX4-expressing rat DRG neurons than CA (*Figure 5c,d*, *Figure 5—figure supplement 1c*). Together, our results indicate that expression of MRGPRX4 is sufficient to render bile acid sensitivity of primary somatosensory neurons.

Next, we asked whether hDRG neurons can also be activated by bile acids. Application of DCA induced a robust fluorescence increase in a subset (~5%) of cultured human embryo DRG neurons loaded with Fluo-8 AM (*Figure 5g,h*, *Figure 5—figure supplement 2a*); the percentage of DCA-responsive cells is similar to the percentage of MRGPRX4-expressing cells measured with in situ hybridization (*Figure 4a,c*). Around 20% of DCA-responsive hDRG neurons were capsaicin-sensitive and histamine-sensitive (*Figure 5i*). The different percentage between the co-response in cultured embryo DRG neurons (*Figure 5i*) and co-expression in adult DRG neurons (*Figure 4f,g*) might be due to the differentiation during development. We also tried to culture adult human DRG neurons derived from schwannoma patients undergoing surgical excision. Both bile acids DCA and CA could trigger a robust calcium response in some cultured cells (*Figure 5—figure supplement 2b1-b3, c1-c3, d1-d3*). The percentage of cells responding to bile acids varied from batch to batch, which may likely be due to the small sample size and different tissue qualities from surgical excision.

## Pharmacological activation of MRGPRX4 triggers itch sensation in human subjects

Given the specific expression pattern of MRGPRX4 in a subset of hDRG neurons, and the known role of Mrgpr family members in mediating itch sensation, we next asked whether pharmacologically activating MRGPRX4 could trigger itch sensation in human subjects. We recruited healthy volunteers and performed a double-blind skin itch test, in which each subject received a 25 µl intradermal injection of the test compounds or vehicle in four separate sites on both forearms (*Figure 6a1*, inset), after which the subject was asked to rank the itch sensation at each injection site using a generalized labeled magnitude scale (LMS) (*Green et al., 1996*). Interestingly, the nateglinide, a FDA approved drug for the treatment of diabetes, previously reported as MRGPRX4 agonist and has itch side effect (*Kroeze et al., 2015*; *Kozlitina et al., 2019*), (*Figure 3—figure supplement 1f*) —but not vehicle—induced a robust itch sensation in healthy subjects (*Figure 6a1, a2*). These results show that activation of MRGPRX4 is sufficient to trigger itch sensation in humans, suggesting that MRGPRX4 is a human itch receptor.

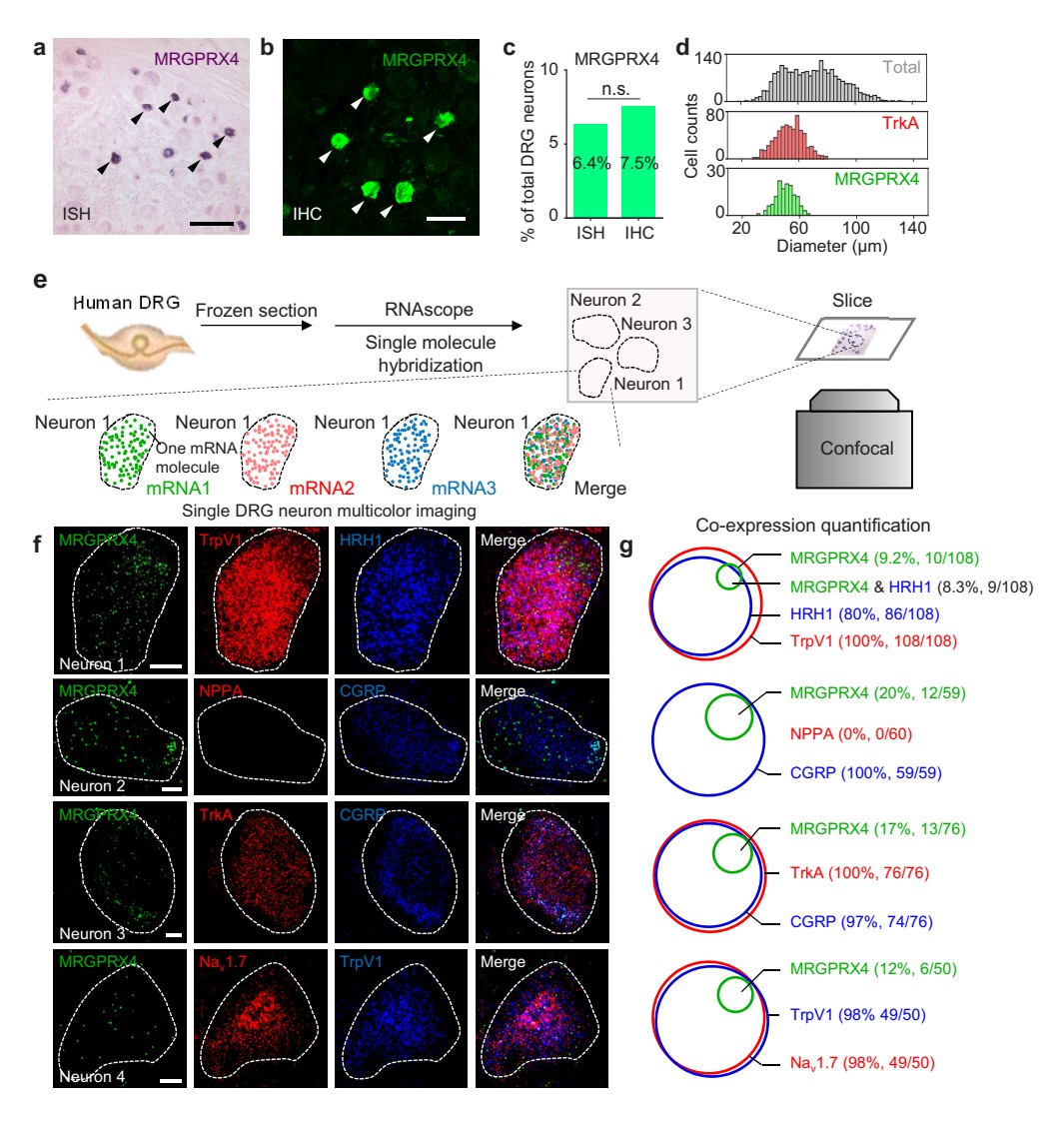

**Figure 4.** MRGPRX4 is expressed in a subset of hDRG neurons. (**a–d**) Representative DRG sections showing in situ hybridization (ISH, (**a**) and immunohistochemistry (IHC, (**b**) for MRGPRX4; the summary data are shown in (**c**); n = 2234 and 2735 neurons for ISH and IHC, respectively. The scale bars represent 200 µm (**a**) and 100 µm (**b**). (**d**) Diameter distribution for all 2234 DRG neurons measured using in situ hybridization, 124 MRGPRX4-positive neurons, and 788 TrkA-positive neurons. Two-proportion z-test, n.s. not significant (p=0.103). (**e**) Flow chart depicting the steps for characterizing the gene expression profiles of human DRG samples using triple-color RNAscope in situ hybridization. (**f**) Representative RNAscope images of *MRGPRX4* and other genes in human DRG sections. Each fluorescent dot indicates a single mRNA transcript. Scale bar, 10 µm. (**g**) Quantification of the gene expression data shown in (**f**). A neuron was defined as positive if ≥ 20 fluorescent dots in the respective mRNA channel were detected in that neuron.

DOI: https://doi.org/10.7554/eLife.48431.008

The following figure supplement is available for figure 4:

**Figure supplement 1.** The anti-MRGPRX4 antibody has high specificity.

DOI: https://doi.org/10.7554/eLife.48431.009

## Bile acid–induced itch in humans is both histamine- and TGR5-independent

Previous studies have implicated that bile acids could induce itch in human (*Kirby et al., 1974*; *Varadi, 1974*). Here, we systematically test pruritic effect of bile acids on human and whether bile acid-

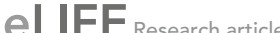

**Figure 5.** MRGPRX4 mediates bile acids induced activation of DRG neurons. (**a**) Top, cultured rat DRG neurons were transfected with the pPiggyBac-CAG-MRGPRX4-P2A-mCherry plasmid by electroporation. The neurons circled by dashed lines are an MRGPRX4-positive neuron (neuron one with red fluorescence) and an MRGPRX4-negative neuron (neuron 2). (**b**) Representative traces from the cells indicated in (**a**). DCA and CA: 10 µM; capsaicin (Cap): 1 µM; KCl: 75 mM. (**c, d**) Summary of the amplitude and percentage of $Ca^{2+}$ signals in response to DCA and CA from MRGPRX4-positive and MRGPRX4 negative neurons in (**a, b**). Responsive neurons were defined as exceeding a threshold of 20% $\Delta F/F_0$. n = 60–77 neurons per group. All error bars represent the s.e.m.. Student's *t*-test and two-proportion z-test, **p<0.01, ***p<0.001. (**e, f**) Representative images and electrophysiological recording of DCA induced action potentials in cultured DRG neurons transfected with MRGPRX4. Cultured rat DRG neurons were transfected with the pPiggyBac-CAG-MRGPRX4-P2A-mCherry plasmid by electroporation. Red fluorescence indicated MRGPRX4-expressing neurons. DCA induced a series of action potentials in MRGPRX4-expressing cells (**e**) but not in non-transfected cells (**f**). Scale bar, 20 µm. (**g**) Bile acids induced a $Ca^{2+}$ response in a subset of cultured human embryo DRG neurons. (left) Representative bright-field images and Fluo-8

*Figure 5 continued on next page*

*Figure 5 continued*

fluorescence images of DRG neuron cultures from one embryo donor. The pseudo-color images show the DCA-induced calcium responses of all imaged neurons. (right) Representative traces of individual DCA-responsive DRG neurons (circled by the dash line in [left]). Pseudo-color images of chemical-induced signals are shown under each trace. C15 (compound 15), CA, DCA, and His (histamine): 100 µM each; KCl: 75 mM. Veh, vehicle. Scale bar, 50 µm. (h) Percentage of human embryo DRG neurons that were responsive to the indicated tested compounds measured as in (g). (i) Venn diagram of the cultured human embryo DRG neurons that were activated by the indicated tested compounds. Green represents DCA responded neurons; Heavy gray represents capsaicin responded neurons; light gray represents histamine responded neurons.

DOI: https://doi.org/10.7554/eLife.48431.010

The following figure supplements are available for figure 5:

**Figure supplement 1.** Expressing MRGPRX4 in cultured rat DRG neurons renders the cells responsive to bile acids.

DOI: https://doi.org/10.7554/eLife.48431.011

**Figure supplement 2.** Cultured human DRG neurons respond to various chemicals.

DOI: https://doi.org/10.7554/eLife.48431.012

induced itch shows some features similar to that of cholestatic itch (*Liu et al., 2012*). We found that 500 µg (25 µl) of DCA induced a significant itch sensation that peaked within 5 min and declined slowly over time; in contrast, control injections with vehicle did not induce an itch response (*Figure 6a1, a2*). Moreover, itch intensity induced by DCA was in a dose-dependent manner (*Figure 6b1, b2*). We also found that less potent MRGPRX4 agonists, including CA, CDCA, tauro-chenodeoxycholic acid (TCDCA), and LCA, also induced a weaker—albeit still significant—itch sensation (*Figure 6c1, c2*). Given that antihistamines are largely ineffective for treating cholestatic itch (*Beuers et al., 2014*), we tested whether itch induced by bile acids can be blocked by antihistamines. We found that pretreating subjects with an antihistamine prevented histamine-induced itch but had no effect on DCA-induced itch (*Figure 6d1, d2*), suggesting that itch induced by bile acids does not involve histamine signaling. Taken together, these results indicate that bile acids trigger an itch sensation with features similar to cholestatic itch.

In mice, the membrane bile acid receptor TGR5 has been reported to mediate bile acid–induced itch (*Alemi et al., 2013*; *Lieu et al., 2014*). To test whether bile acid-induced itch in human is also mediated by TGR5, we chose a non-bile acid TGR5 agonist compound 15 (*Högenauer et al., 2014*), which is nearly 70-fold more potent than DCA in activating human TGR5 and does not activate human MRGPRX4 (*Figure 7b,c*). Intradermal injections of 10 µg (25 µl) of compound 15 did not induce detectable itch in humans, whereas DCA, as the positive control, induced significant itch (*Figure 7a1, a2, d*). These results suggest that TRG5 is not the receptor mediating bile acid-induced itch. Furthermore, we examined the expression of TGR5 in the human, monkey, and mouse DRG tissues. Very surprisingly, although the amino acid sequence of TGR5 is relatively conserved between rodents and primates (*Figure 7—figure supplement 1a*), we found the different expression pattern of TGR5 in DRG tissues. In human and monkey, both in situ hybridization and immunostaining revealed that TGR5 is highly expressed in satellite glial cells surrounding DRG neurons but not the primary sensory neurons (*Figure 7e,g* and *Figure 7—figure supplement 1d,e*), while in mouse, the same in situ probe and antibody detected the expression of TGR5 in mouse DRG neurons (*Figure 7f,h* and *Figure 7—figure supplement 1c*), similar to the previous publication (*Alemi et al., 2013*; *Lieu et al., 2014*). Our results revealed an interesting species difference in TGR5 expression and function between mouse and primate. Taken together, our results demonstrate that the function of TGR5 in human somatosensory system is different from that in mouse, and TGR5 is not the receptor for mediating bile acid-induced itch in human.

## The elevated levels of bile acids in cholestatic itchy patients are sufficient to activate MRGPRX4

Lastly, to investigate whether bile acids are the pruritogens under pathological conditions, we collected plasma samples from patients with liver or skin diseases and measured the concentration of 12 major bile acids using HPLC-MS/MS (*Figure 8a* and *Figure 8—figure supplement 1a*). We found that glycine- and taurine-conjugated primary bile acids, including glycocholic acid (GCA), taurocholic

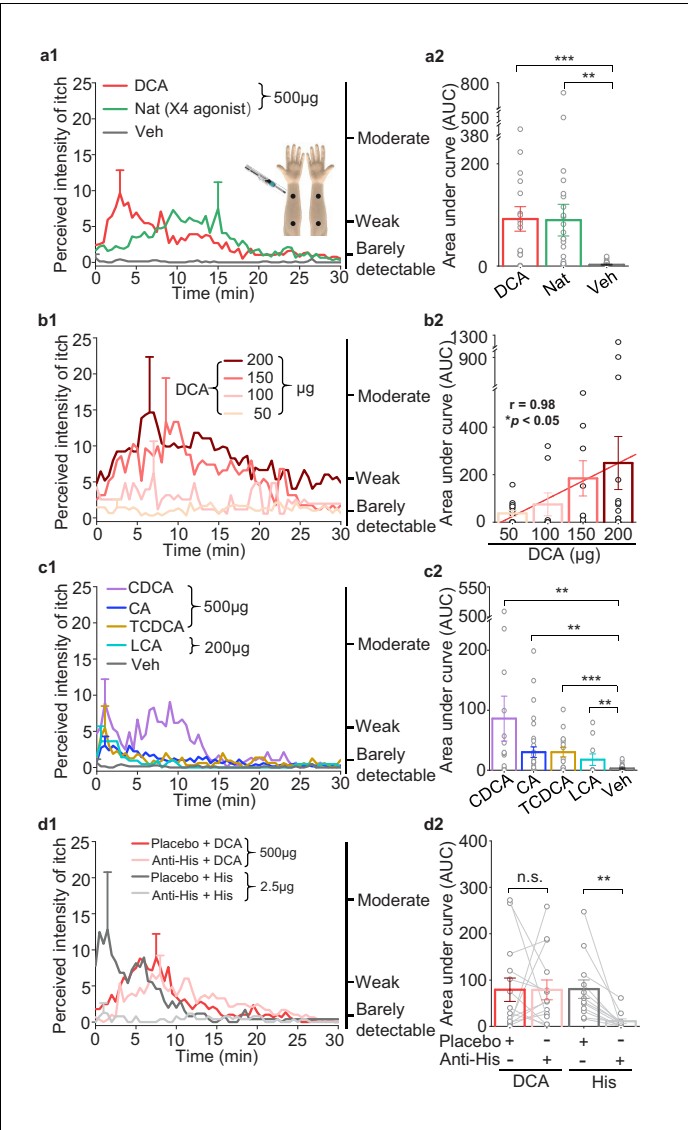

**Figure 6.** Bile acids and MRGPRX4 specific agonist induce histamine-independent itch in human. (**a1–a2**) Itch evoked by a double-blind intradermal injection of DCA and nateglinide (Nat) in human subjects. (25 µl for each injection) (**a1**) Time courses of the perceived itch intensity (n = 18–32. The traces are plotted with the standard error of the mean (s.e.m.) at the peak of each trace. The descriptions of the itch intensity are shown on the right. The injection sites on the subject's forearm are indicated (inset). X4, MRGPRX4 (**a2**) Summary of the area-under-curve (AUC) of the itch intensity traces shown in (**a1**). (**b1–b2**) Itch evoked by the indicated doses of DCA (25 µl for each injection, n = 8–14). The linear regression analysis of concentration versus the AUC is showed as a red line. (**c1–c2**) Itch evoked by CDCA, CA, TCDCA, and LCA (25 µl for each injection, n = 10–31). The vehicle data (Veh) is reproduced from (**a1**). (**d1–d2**) DCA-evoked itch is not inhibited by antihistamine (Anti-His). (**d1**) Time course of itch intensity evoked by an intradermal injection of DCA or histamine (His) following antihistamine or placebo pretreatment (25 µl for each injection, n = 12–14). Each pair of dots connected by a gray line represents an individual subject. All error bars represent the s.e.m.. Student's t-test, *p<0.05, **p<0.01, ***p<0.001, and n.s. not significant (p>0.05).

DOI: https://doi.org/10.7554/eLife.48431.013

acid (TCA), glycochenodeoxycholic acid (GCDCA), and TCDCA are the major bile acids present in cholestatic patients (*Figure 8a*), consistent with previously published results (*Neale et al., 1971*; *Freedman et al., 1981*; *Bartholomew et al., 1982*). Compared to non-itchy patients with liver diseases, itchy patients had significantly higher levels of total bile acids (defined here as the sum of the

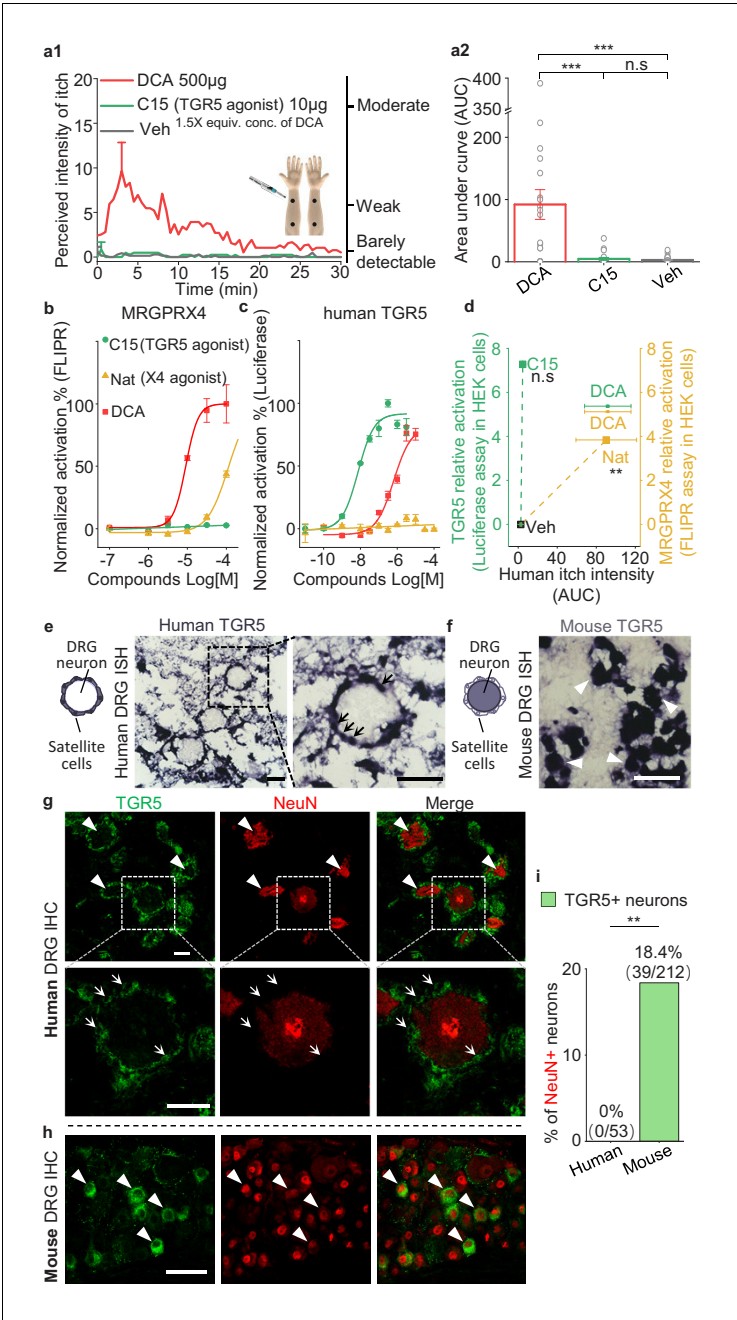

**Figure 7.** TGR5 does not serve as an itch receptor in human. (**a1–a2**) Intradermal injection of a non-bile acid TGR5 agonist compound 15 (C15) does not induce itch in human. (**a1**) Time courses of the perceived intensity of itch evoked by DCA and vehicle are reproduced from *Figure 5a1*, and the itch evoked by C15 is from 19 subjects. The equivalent concentration (equiv. conc.) of DCA and C15 means the fold of concentration to the $EC_{50}$ of activating human TGR5. (**a2**) The quantification results of area under curve (AUC) of itch intensity shown in (**a1**) (mean ± s.e. m.). Veh, vehicle. Student's *t*-test, ***p<0.001, and n.s. not significant (p>0.05). (**b–c**) The activation of human MRGPRX4 (**b**) or human TGR5 (**c**) by DCA (red), compound 15 (C15, green) and nateglinide in MRGPRX4- or TGR5-expressing HEK293T cells detected by FLIPR and luciferase assay respectively. (**d**) The relationship between the evoked itch and the relative potency to activate human MRGPRX4 or human TGR5 by the specific agonists of these two receptors. The Y-axis shows the relative activation of certain compound to the receptor, representing the logarithm of (maximal response/$EC_{50}$). The X-axis shows the human itch intensity, representing the AUC of itch evoked by certain compound. Statistic test was performed between the itch intensity of compound 15 and vehicle, or between the itch intensity of nateglinide and vehicle. Nat, nateglinide; C15, compound 15; Veh, vehicle. Student's *t*-test, **p<0.01, and n.s. not significant (p>0.05) versus vehicle treatments. (**e**) In situ hybridization (ISH)

*Figure 7 continued on next page*

*Figure 7 continued*

of TGR5 in human DRG sections. (left) The diagram depicting the morphology of DRG neurons and surrounding satellite glial cells. (middle and right) TGR5 was highly expressed in satellite glial cells (indicated by arrows) but not DRG neurons in human DRG. Scale bar, 50 µm. (f) In situ hybridization of TGR5 in mouse DRG sections. TGR5 was highly expressed in DRG neurons (indicated by arrow heads) in mouse DRG. Scale bar, 50 µm. (g–h) Immunohistochemistry (IHC) of human and mouse DRG sections. (g) In human DRG, TGR5 was expressed in satellite glial cells (indicated by arrows) but not in neurons (marked by NeuN, indicated by arrow heads). (h) In mouse DRG, TGR5 was expressed in neurons (marked by NeuN, indicated by arrow heads). Scale bar, 50 µm. (i) Quantification of the percentage of TGR5+ neurons (over NeuN+ neurons) in human and mouse DRG (immunohistochemistry). Chi-square test, **$p < 0.01$. All error bars represent the s.e.m.

DOI: https://doi.org/10.7554/eLife.48431.014

The following figure supplement is available for figure 7:

**Figure supplement 1.** Expression of TGR5 in mouse and monkey DRG.

DOI: https://doi.org/10.7554/eLife.48431.015

---

12 bile acids shown in *Figure 8a*) (*Figure 8a,b*). The level of total plasma bile acids in the itchy patients with skin diseases was barely detectable and significantly lower than the itchy patients with liver diseases. Among the 12 bile acids measured, the ones with the largest differences between the patients with itch and those without itch were for GCA, TCA, GCDCA, and TCDCA (*Figure 8a,b*), suggesting that these four bile acids play key roles in mediating chronic itch under pathological conditions. Indeed, intradermal injections of TCDCA caused significant itch in healthy subjects (*Figure 6c1, c2*). For DCA, the most potent ligand for MRGPRX4 among all tested bile acids, we did not see the significant difference between itchy and non-itchy patients with liver diseases (*Figure 8a* and *Figure 8—figure supplement 1b*), suggesting it is not the major contributor for cholestatic itch under pathological conditions. More importantly, although bile acid levels vary among itchy patients with liver diseases both from our data (*Figure 8a,b*) and previously reported results (*Freedman et al., 1981*; *Bartholomew et al., 1982*; *Schoenfield and Sjovall, 1967*), we found that the total plasma bile acids, as well as the individual levels of GCA, TCA, GCDCA, and TCDCA significantly decreased in 11 out of 13 patients following itch relief (*Figure 8c,d* and *Figure 8—figure supplement 1c*). Taken together, these results suggest that high levels of bile acids are well correlated with itchy symptom in patients with liver diseases and that bile acids—particularly GCA, TCA, GCDCA, and TCDCA— could be main metabolites triggering cholestatic itch.

Next, we examined whether combinations of bile acids at pathologically relevant levels are sufficient to activate MRGPRX4. We prepared mixtures of bile acids similar to the plasma/serum levels in healthy subjects ('healthy mix') or in patients with liver diseases and itch ('liver_itch mix'), which are estimated based on previously published data (*Neale et al., 1971*; *Xiang et al., 2010*) and our quantification results (*Figure 8a*). These mixtures were then applied to MRGPRX4-expressing HEK293T cells while performing $Ca^{2+}$ imaging. We found that the 'liver_itch mix' but not 'healthy mix' induced a significant $Ca^{2+}$ signal (*Figure 8e,f*), suggesting that pathological relevant level of bile acids is sufficient to activate MRGPRX4.

Recently, Meixiong et al. reported that MRGPRX4 can also be activated by bilirubin, which is another potential pruritogen for triggering cholestatic itch (*Meixiong et al., 2019a*). We therefore compared bilirubin and DCA with respect to binding and activating MRGPRX4 in HEK293T cells. We found that compared to bile acids, bilirubin is a less potent, partial agonist of MRGPRX4 (*Figure 9a*). Given the structural differences between bilirubin and DCA, we then tested whether bilirubin is an allosteric modulator of MRGPRX4. Indeed, we found that bilirubin can potentiate the activation of MRGPRX4 by DCA (*Figure 9b*), and—conversely—DCA potentiate the activation of MRGPRX4 by bilirubin (*Figure 9c*). In addition, we also compared the activity of bilirubin and DCA to MRGPRX4 in cultured rat DRG neurons. Similar to the results in HEK293T cells, DCA induced larger $Ca^{2+}$ responses in MRGPRX4-expressing neurons compared to bilirubin, and the DCA-responsive neurons was about 2.5-fold more than bilirubin-responsive neurons (*Figure 9d–g*). Moreover, we found that both total bilirubin and conjugated bilirubin levels were significantly higher in itchy patients with liver diseases compared to non-itchy patients (*Figure 9h*), and plasma bilirubin levels decreased significantly after itch relief (*Figure 9i*). Compare to total bilirubin, total bile acids show better correlation with itch intensity (measured using a self-report numerical rating scale; *Jenkins et al., 2009*;

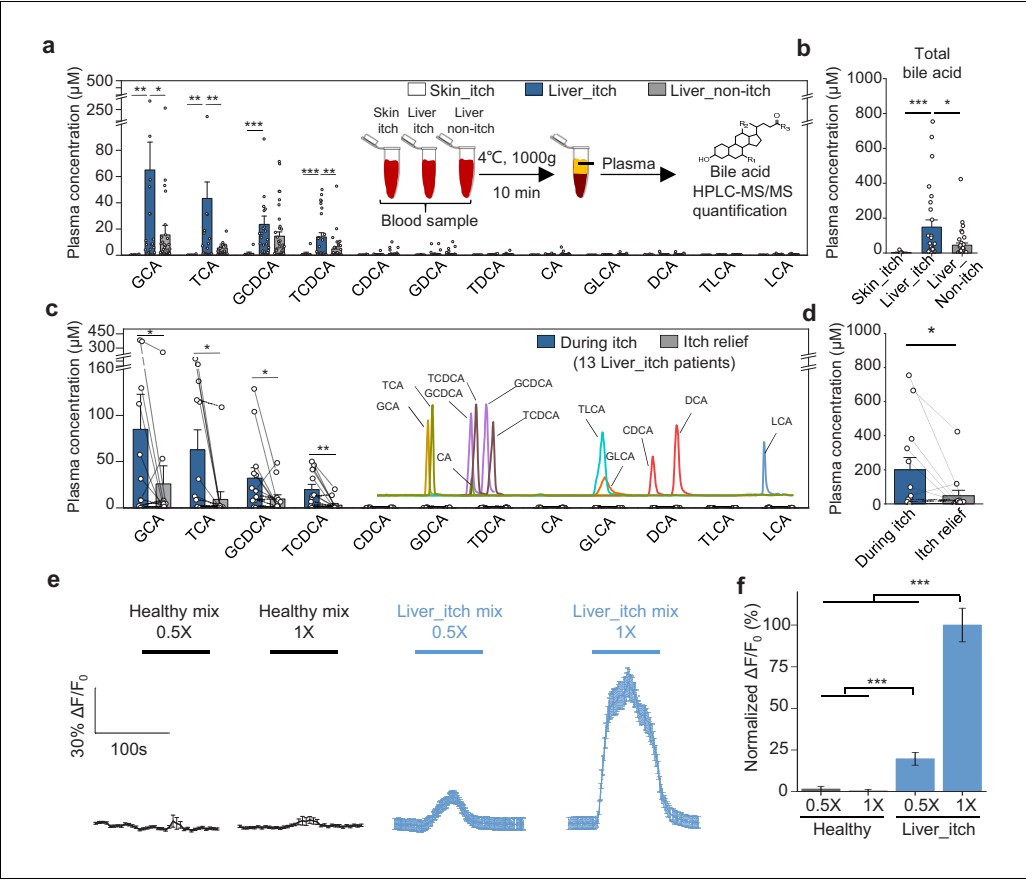

**Figure 8.** Elevated bile acids are correlated with the occurrence of itch among patients with liver disease and are sufficient to activate MRGPRX4. (**a–b**) Summary of individual bile acid levels (**a**) and total bile acid levels (**b**), the sum of the 12 bile acids shown in a) in itchy patients with liver diseases (Liver_itch, n = 27), non-itchy patients liver with diseases, (Liver_non-itch, n = 36), and itchy patients with dermatic diseases (Skin_itch, n = 8). The plasma bile acid levels were measured using HPLC-MS/MS (inset). (**c–d**) Summary of individual bile acid levels (**c**) and total bile acid levels (**d**), the sum of the 12 bile acids shown in c) in 13 patients with liver diseases during itch and after itch relief. The inset shows the separation of standard bile acids by HPLC-MS/MS. (**e–f**) Left, $Ca^{2+}$ responses in MRGPRX4-expressing HEK293T cells induced by application of a mixture of artificial bile acids derived from itchy patients with liver diseases and healthy subjects. The $Ca^{2+}$ signal was measured using Fluo-8 and was normalized to the signal measured using the 1x liver_itch mix. The summary data are shown in (**f**); n = 50 cells each. All error bars represent the s.e.m.. Student's *t*-test, *p<0.05, **p<0.01, ***p<0.001.
DOI: https://doi.org/10.7554/eLife.48431.016

The following figure supplement is available for figure 8:

**Figure supplement 1.** Quantification of bile acids in human plasma.
DOI: https://doi.org/10.7554/eLife.48431.017

*Figure 9j*). Taken together, these results suggest that bile acids are the major pruritogens in MRGPRX4-mediated cholestatic itch, while bilirubin facilitates the activation of MRGPRX4 by bile acids and may also contribute to cholestatic itch in pathological conditions.

## Discussion

In this study, we report that MRGPRX4 is a novel GPCR that fits with the criteria we set for identifying putative receptor in mediating cholestatic itch in human. MRGPRX4 is selectively expressed in a small subset of human DRG neurons. Bile acids triggered a robust $Ca^{2+}$ response in a subset of hDRG neurons as well as rat DRG neurons expressing MRGPRX4 exogenously. Both bile acids and an MRGPRX4-specific agonist induced itch in human. Bile acid-induced itch in human wass histamine

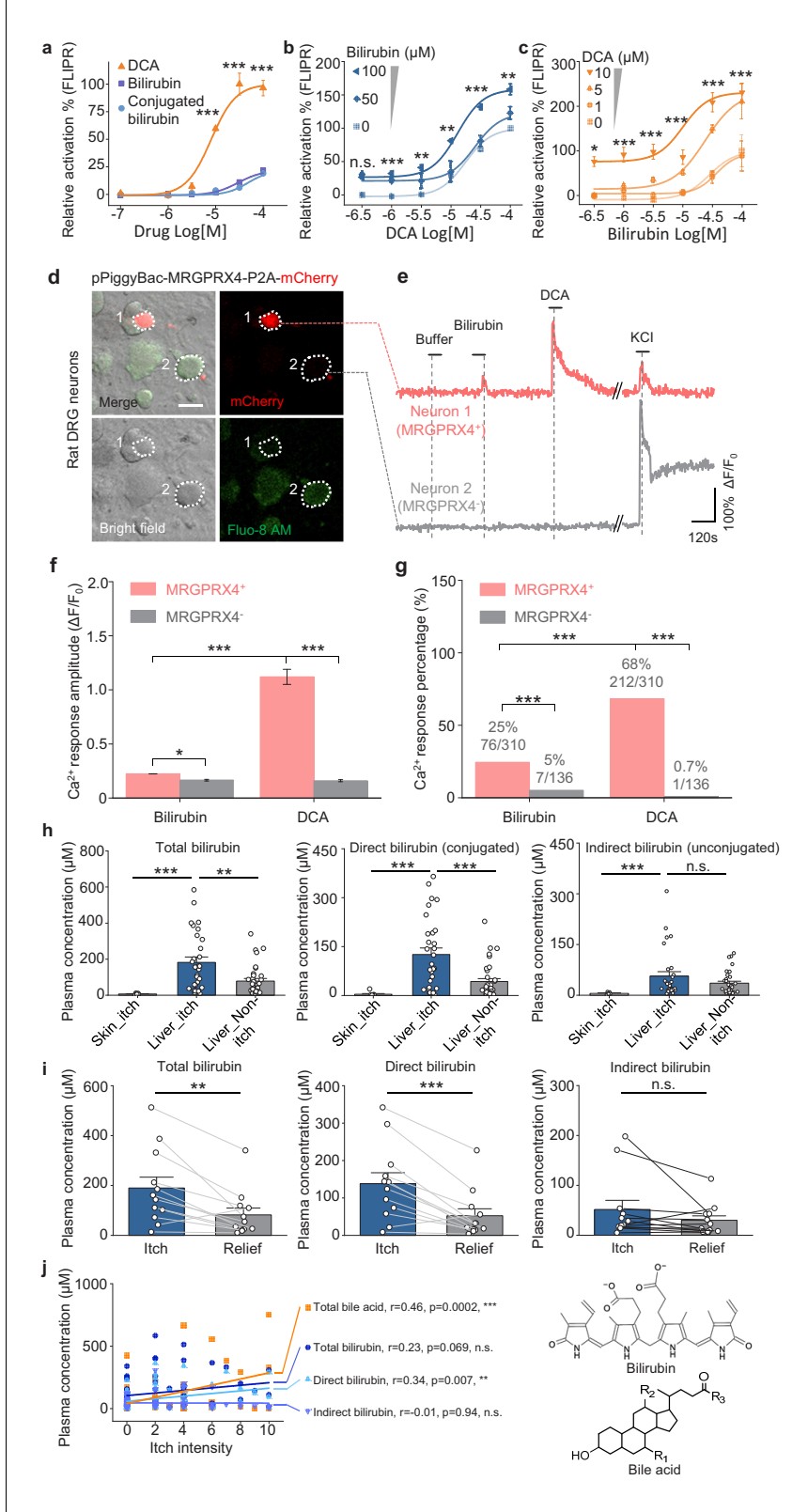

**Figure 9.** Bilirubin potentiates the activation of MRGPRX4 by bile acids and may contribute to cholestatic itch. (a) Comparison of the activation of MRGPRX4 by DCA, bilirubin and taurine conjugated bilirubin. Taurine conjugated bilirubin was used in order to mimic the direct bilirubin under human physiological condition. MRGPRX4 was expressed in HEK293T cells and the activation was measured by FLIPR assay. (b) Bilirubin allosterically modulates

*Figure 9 continued on next page*

*Figure 9 continued*

the activation of MRGPRX4 by DCA. Different concentrations of bilirubin was mixed with DCA, and then the activation of MRGPRX4 by these mixes was tested in MRGPRX4-expressing HEK293T cells using FLIPR assay. (c) DCA allosterically modulates the activation of MRGPRX4 by bilirubin, similar to (b). (d) Cultured rat DRG neurons were transfected with the PiggyBac-CAG-MRGPRX4-P2A-mCherry plasmids by electroporation. The neurons circled by dashed lines are an MRGPRX4-positive neuron (neuron one with red fluorescence) and an MRGPRX4-negative neuron (neuron 2). Scale bar, 20 μm. (e) Representative traces from the cells indicated in (d). Bilirubin and DCA: 100 μM; KCl: 75 mM. (f–g) Summary of the amplitude (f) and percentage (g) of $Ca^{2+}$ signals in response to bilirubin and DCA. Responsive neurons were defined as exceeding a threshold of 10% $\Delta F/F_0$. n = 136–310 neurons per group. (h) Comparison of total bilirubin, direct bilirubin (conjugated) and indirect bilirubin (unconjugated) level in liver disease patients with itch (Liver_itch) (n = 30) or without itch (Liver_Non-itch) (n = 34), or patients with dermatic itch (Skin_itch) (n = 6). (i) Comparison of total bilirubin, direct bilirubin and indirect bilirubin level in liver disease patients (n = 12) during itch and after itch relief. (j) Correlation between itch intensity and plasma total bile acid, total bilirubin, direct bilirubin, and indirect bilirubin. The itch intensity was directly reported by patients via a questionnaire with 0 representing no itch and 10 the highest level of itch. All error bars represent the s.e.m.. (a-c) One-way ANOVA, *p<0.05, **p<0.01, ***p<0.001, and n.s. not significant (p>0.05). (f-j) Student's *t*-test and two proportion z-test, *p<0.01, **p<0.01, ***p<0.001, and n.s. not significant (p>0.05).

DOI: https://doi.org/10.7554/eLife.48431.018

independent, which is consistent with that antihistamines are largely ineffective for treating cholestatic itch. Surprisingly, application of agonist for TGR5 failed to elicit $Ca^{2+}$ response in cultured hDRG neurons, nor did it induce pruritus in human subjects. More interestingly, we found that the expression pattern of TGR5 was different between mouse and human: hTGR5 was selectively expressed in satellite glial cells, while mTGR5 was expressed in DRG neurons, likely accounting for the inter-species difference functionally. We also found that plasma levels of bile acids were well correlated with the itch intensity of itchy patients with liver diseases. Importantly, a mixture of bile acids with components and concentrations similar to that of cholestatic itchy patients—but not healthy volunteers—was sufficient to activate MRGPRX4. Our data indicate that bile acids are the major pruritogens in MRGPRX4-mediated cholestatic itch, while bilirubin facilitates the activation of MRGPRX4 by bile acids and may also contribute to cholestatic itch in pathological conditions.

Based on all these evidences, we propose a new working model for cholestatic itch (*Figure 10*): patients with cholestasis usually display increased plasma levels of bile acids and bilirubin, which are precipitated in the skin and activate MRGPRX4 in itch-related primary fibers, thereby triggering itch in these patients. Our results exclude TGR5 as a primary itch receptor in human, and the broad expression of TGR5 in satellite glial cells implies a more general function which remains to be determined in the future.

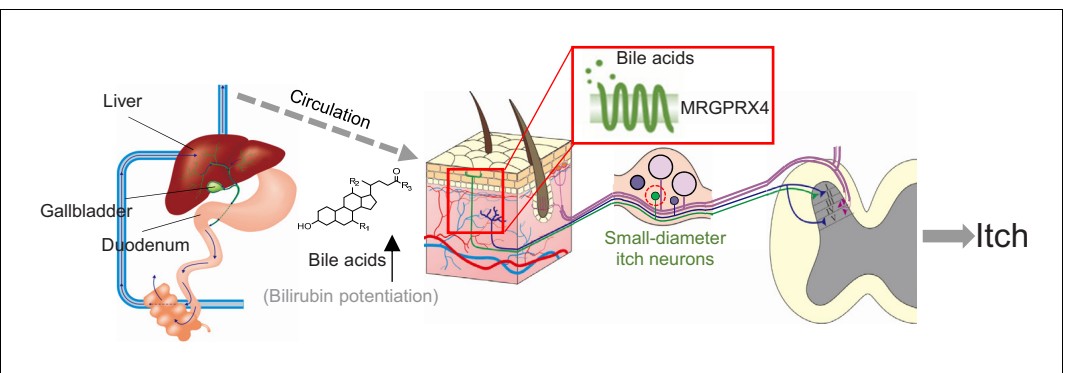

**Figure 10.** Proposed model depicting the mechanism underlying itch in patients with liver diseases. In itchy patients, accumulated bile acids reach the skin via the circulatory system, where they activate nerve fibers of a subset of MRGPRX4-expressing DRG neurons. Bilirubin, as a partial agonist, facilitates the activation of MRGPRX4 by bile acids. These activated neurons relay the itch signal to the spinal cord and higher brain centers, eliciting the sensation of itch.

DOI: https://doi.org/10.7554/eLife.48431.019

Recently, Meixiong et al. reported that and human MRGPRX4 can be activated by bilirubin and bile acids, suggesting that bilirubin and bile acids may serve as a pruritogen in cholestatic itch (*Meixiong et al., 2019a*). Bilirubin, a yellow compound that causes the yellow discoloration in jaundice, has not been considered a likely candidate pruritogen though, because the clinical observations find that itch often precedes the appearance of jaundice, particularly in patients with intrahepatic cholestasis of pregnancy (ICP) (*Geenes and Williamson, 2009*) and patients with primary biliary cirrhosis (*Kremer et al., 2011*). Our results suggest that bile acids is full agonist of MRGPRX4, and bilirubin is a partial agonist potentiating the activation of MRGPRX4 by bile acids. This notion is consistent with our finding that the correlation between bile acid levels and itch intensity is stronger than the correlation between bilirubin levels and itch intensity. Based on these findings, we propose that bile acid is the major contributor to cholestatic itch, and bilirubin serves to increase bile acid–induced cholestatic itch under pathological conditions.

Here, we provide important evidence that MRGPRX4 is sufficient for mediating bile acid–induced itch, and thus should play an important role in cholestatic itch. Since specific antagonist for MRGPRX4 is currently unavailable, we could not determine whether MRGPRX4 is necessary for bile acids induced itch in human. Future studies will be designed to further examine the role of MRGPRX4 in cholestatic itch using to-be-developed pharmacological and/or human genetic approaches. For example, several single-nucleotide polymorphisms (SNPs) have been identified in the human *MRGPRX4* gene (*Lek et al., 2016*), and it would be interesting to screen for loss-of-function and gain-of-function *MRGPRX4* variants. Characterizing the correlation between these functional variants or expression level of MRGPRX4 in DRG neurons with itch intensity in cholestatic patients and healthy subjects with bile acid–induced itch could help to further delineate the relationship between MRGPRX4 activity and cholestatic itch and explain the observed discrepancy between bile acids levels and cholestatic itch intensity (*Beuers et al., 2014*; *Düll and Kremer, 2018*). These experiments will also help to determine whether MRGPRX4 is the main molecular receptor for mediating cholestatic itch, or whether other GPCRs, such as lysophosphatidic acid receptors and serotonin receptors (*Beuers et al., 2014*), and unidentified receptors also play roles in cholestatic itch.

Our current understandings about mechanisms underlying somatosensation in the mammalian system are mainly derived from studies of rodents. Despite the great value and insights we gained using rodent models, notable failures have happened in translating results obtained in rodents into effective and safe clinical treatments in human (*Hill, 2000*; *Mogil, 2009*; *Hug and Weidner, 2012*; *Taneja et al., 2012*). The bile acid receptors we study here is a great example demonstrating the species differences between rodent and human somatosensory systems. Although TGR5, a bile acid membrane receptor, was previously reported to be expressed in mouse DRG neurons and mediate bile acid–induced itch in mice (*Alemi et al., 2013*; *Lieu et al., 2014*), our expressing characterizations as well as functional assays revealed that TGR5 is not expressed in human DRG neurons and doesn't directly mediate itch sensation in human. Instead, primate MRGPRX4 gains the novel function of bile acid sensitivity during evolution. Therefore, it is crucial to study and validate the mechanism of cholestatic chronic itch and develop the correspondent treatment within the context of human physiology.

In summary, we found that the membrane-bound GPCR MRGPRX4 is a novel bile acid receptor and may serve as an important molecular mediator of chronic itch in patients with systemic liver diseases. Our results suggest that MRGPRX4 is a promising molecular target for developing new treatments to alleviate devastating chronic itch in these patients.

## Materials and methods

**Key resources table**

| Reagent type (species) or resource | Designation | Source or reference | Identifiers | Additional information |
|---|---|---|---|---|
| Gene (*H. sapiens*) | MRGPRX4 | hORFeome Database | Genbank Accession: KJ900052 | |

*Continued on next page*

*Continued*

| Reagent type (species) or resource | Designation | Source or reference | Identifiers | Additional information |
|---|---|---|---|---|
| Gene (*H. sapiens*) | TGR5 | hORFeome Database | Genbank Accession: KJ895655 | |
| Cell line (*Homo-sapiens*) | HEK stable cell line for TGFα shedding assay | PMID: 22983457 | | Self-generated according to reference paper |
| Chemical compound, drug | Screen Quest Fluo-8 No Wash Calcium Assay Kit | AAT Bioquest | Cat. #: 36316 | |
| Chemical compound, drug | Fluo-8, AM | AAT Bioquest | Cat. #: 21083 | |
| Chemical compound, drug | Deoxycholic acid | Sigma-Aldrich | Cat. #: D2510 | |
| Chemical compound, drug | U73122 | Selleck | Cat. #: S8011 | |
| Chemical compound, drug | Nateglinide | J and K Scientific | Cat. #: 586681 | |
| Chemical compound, drug | Bilirubin ditaurate | J and K Scientific | Cat. #: F-H130 | |
| Chemical compound, drug | Compound 15 | custom- synthesized | | Verified by HPLC and mass spectrum |
| Chemical compound, drug | Histamine | Sigma-Aldrich | Cat. #: H7250 | |
| Antibody | Rabbit polyclonal anti-hMRGPRX4 | Abcam | Cat #: ab120808 | IHC (1:200) |
| Antibody | Rabbit polyclonal anti-hTGR5 | Thermofisher | Cat #: PA5-27076 RRID:AB_2544552 | IHC (1:200-1:500) |
| Antibody | Mouse monoclonal anti-NeuN (clone A60) | Sigma-Aldrich | Cat. #: MAB377 RRID:AB_2298772 | IHC (1:1000) |
| Antibody | Rabbit polyclonal anti-c-Myc | Sigma-Aldrich | Cat. #: C3956 RRID: AB_439680 | IHC (1:200) FACS (1:50) |
| Antibody | Sheep polyclonal anti-Digoxigenin-AP | Roche | Cat. #: 11093274910 RRID:AB_2734716 | ISH (1:1000) |
| Commercial assay or kit | RNAscope Fluorescent Multiplex Assay | Advanced Cell Diagnostics | Cat. #: 320293 UM | |
| Commercial assay or kit | P3 Primary Cell 4D-NucleofectorTM X Kit L | Lonza | Cat. # V4XP-3012 | |
| Recombinant DNA reagent | pPiggyBac-MRGPRX4-P2A-mCherry (Plasmid) | This paper | | See in *Figure 1—figure supplement 1a* |
| Recombinant DNA reagent | pPiggyBac-TGR5-P2A-mCherry (Plasmid) | This paper | | See in *Figure 1—figure supplement 1a* |
| Other | Doxepin hydrochloride cream | Chongqing Huapont Pharm. Co. | | |
| Other | Cold cream | Eau Thermale Avène | | |

## Analysis of GPCRs expressed in human DRG neurons

The expression profile of all genes in hDRG neurons was compared to human reference tissues, including trigeminal ganglia, brain, colon, liver, lung, muscle, and testis (*Flegel et al., 2013*; *Flegel et al., 2015*). To identify DRG-enriched GPCRs, we using the following formula: [(the expression level of a given gene in the DRG)/(the total expression level of that gene in all tissues)]; a value $\geq$ 0.5 was used to define DRG-enriched genes. The expression level of a gene refers to the number of fragments per kilobase of exon per million fragments mapped (FPKM) in the tissue transcriptome.

## Bovine tissue extracts

Fresh bovine heart, brain, kidney, spleen, and liver tissues (40 g each) were dissected and then boiled for 5 min in 200 ml water. Acetic acid and HCl were then added to a final concentration of 1 M and 20 mM, respectively, and the mixture was homogenized thoroughly and then centrifuged at 11,000 rpm for 30 min. The supernatant was collected and concentrated to a volume of 40 ml using a rotary evaporator. Acetone (80 ml) was then added to the concentrated solution, and the new solution was again centrifuged at 11,000 rpm for 30 min. The supernatant was collected using a rotary evaporator and freeze-dried in a vacuum. The final product was weighed, and equal amounts of each extract were used to test for activity.

## Generation of stable GPCR-expressing cell lines

Stable cell lines expressing human DRG-enriched candidate GPCRs were generated using the Piggy-Bac Transposon System. In brief, each GPCR was subcloned into the PiggyBac Transposon vector and co-transfected with the hyperactive PiggyBac transposase (*Yusa et al., 2011*) into the HEK293T-based TGFα shedding reporter cell line (*Inoue et al., 2012*) using polyethylenimine (PEI). Receptor-expressing cells were selected and maintained in DMEM containing 10% fetal bovine serum (FBS), 1 μg/ml puromycin, 100 U penicillin, and 100 μg/ml streptomycin in a humidified atmosphere at 37°C containing 5% $CO_2$.

## TGFα shedding assay

Cultured cells expressing candidate GPCRs were rinsed once with $Mg^{2+}$-free and $Ca^{2+}$-free phosphate-buffered saline (PBS) and then detached with 0.05% (w/v) trypsin. The cell suspension was transferred to a 15 ml tube and centrifuged at 190x*g* for 5 min. The supernatant was discarded, and the cell pellet was suspended in 10 ml PBS and incubated for 15 min at room temperature (RT). The cells were re-centrifuged and suspended in 4 ml HBSS (Hanks' balanced salt solution) containing 5 mM HEPES (pH 7.4). The suspended cells were then seeded in a 96-well plate at 40,000–50,000 cells per well and placed in a 37°C incubator in 5% $CO_2$ for 30 min. A 10x stock solution of each drug was prepared in assay buffer (HBSS containing 5 mM HEPES, pH 7.4), and 10 μl of 10x stock solution was added to each well. The plate was then placed in the incubator for 2 hr, after which alkaline phosphatase (AP) activity was measured in the conditioned media and cells.

## FLIPR assay

HEK293T cells stably expressing human MRGPRX4 were seeded in 96-well plates at a density of ~ 50,000 cells per well. The following day, the cells were loaded with Fluo-8 (Screen Quest Fluo-8 No-Wash Calcium Assay Kit, AAT Bioquest, cat. # 36316) for 2 hr, and test compounds were added to the wells. The Fluo-8 signal was measured using the FLIPR TETRA system (PerkinElmer).

## Luciferase assay

We generated a luciferase reporter plasmid that encodes secreted NanoLuc under the control of a cAMP response element (CRE) and a minimal promoter. The hygromycin-resistance gene and EBFP driven by the SV40 promoter in the reporter plasmid were used to generate stable cell lines. HEK293T cells were transfected with this plasmid, and a stable cell line was generated by selecting with hygromycin.

This stable reporter cell line was then transfected with various GPCRs and used to monitor the activation of these receptors. In brief, the cells were seeded in 96-well plates; the next day, the culture medium was replaced, and compounds were added to the wells; forskolin (10 μM final concentration) and 0.01% DMSO (v/v) were used as positive and negative controls, respectively. The plates were incubated at 37°C in 5% $CO_2$ for 24 hr, after which a 10 μl aliquot of cell culture medium was removed from each well and combined with 40 μl culture medium plus 50 μl assay buffer (containing 20 μM of the luciferase substrate coelenterazine); after 5 min incubation, luminescence was measured using an EnVision plate reader (PerkinElmer).

## Fractionation of bile acid components

A commercially available bovine bile acid powder (126.6 mg) was loaded in a silica gel column (DCM:MeOH = 10:1). The smaller fractions were combined to form six larger fractions (F1 through

F6) based on analytical thin-layer chromatography performed using 0.25 mm silica gel 60 F plates. Flash chromatography was performed using 200–400 mesh silica gel.

## MS and NMR

High-resolution mass spectrometry was performed at the Peking University Mass Spectrometry Laboratory using a Bruker Fourier Transform Ion Cyclotron Resonance Mass Spectrometer Solarix XR. $^1$H-NMR spectra were recorded on a Bruker 400-MHz spectrometer at ambient temperature with $CDCl_3$ as the solvent.

## Immunostaining and flow cytometry analysis

Suspended live HEK 293 cells stably expressing the point-mutated MRGPRX4 were washed in washing buffer (1X PBS solution, mixed with 5% fatal bovine serum [FBS]) for 3 times. Then cells were incubated with the rabbit anti-c-Myc primary antibody (Sigma-Aldrich cat. # C3956, 1:25 dilution) for 30 min, and secondary antibody (AAT Bioquest iFluro Alexa 488 goat antirabbit IgG, cat. # 1060423, 1:50 dilution) for 1 hr. Cells were washed for two times after each antibody treatment. Next, cells were resuspended with 300 uL to 500 uL FACS buffer, and fluorescence-activated cell-sorting analysis was performed, using the BD FACS Calibor Flow cytometer (BD Biosciences), and the data were analyzed using FlowJo software (Ver. 7.6.1).

## Cultured human DRG neurons

Collection of DRG tissue from adult humans was approved by the Committee for Medical Science Research Ethics, Peking University Third Hospital (IRB00006761-2015238), and collection from human embryos was approved by the Reproductive Study Ethics Committee of Peking University Third Hospital (2012SZ-013 and 2017SZ-043) and Beijing Anzhen Hospital (2014012x). DRG tissues were obtained from adult patients undergoing surgical excision of a schwannoma; the tissues were placed immediately in ice-cold DMEM/F12 medium. The tissues were then cut into pieces < 1 mm in size and treated with an enzyme solution containing 5 mg/ml dispase and 1 mg/ml collagenase at 37°C for 1 hr. After trituration and centrifugation, the cells were washed in 15% (w/v) bovine serum albumin (BSA) resuspended in DMEM/F12 containing 10% FBS, plated on glass coverslips coated with poly-D-lysine and laminin, cultured in an incubator at 37°C, and used within 24 hr of plating.

## Electrophysiology

Culture and electroporation of rodent DRG neurons were performed as previous described. The borosilicate glass electrodes (Sutter Instrument) were pulled to a tip resistance of 2.5–5 MΩ and filled with internal solution containing 145 mM potassium gluconate, 5 mM $MgCl_2$, 10 mM HEPES, and 4 mM $Na_2$-ATP (pH 7.2, with KOH). The glass electrode's position was adjusted by a Sutter MP285 micro-manipulator. Membrane potentials were recorded under whole-cell current clamp under I = 0 mode (Axopatch 200B, Axon Instruments). For evaluating intrinsic excitability, DRG neurons were stimulated with stepwise current injection from 5 pA to 100 pA or 110 pA to 300 pA in 20 steps with 473 ms stimulation and 473 ms rest in each step (at 21159.48 Hz) under I-CLAMP NORMAL mode. Recorded membrane potential data were filtered with a 5 kHz internal Bessel filter in the amplified and digitized with a National Instruments PCIe-6353 data acquisition (DAQ) board. The microscope (Nikon Ti-E), the camera (Hamamatsu ORCA-Flash 4.0 v2) and electrophysiology recording system were controlled with a customized software written in LabVIEW (National Instruments), and the data were extracted and analyzed with a home-made software written in MATLAB (MathWorks).

## Culture and electroporation of rodent DRG neurons

Rat DRG tissues were obtained from the thoracic and lumbar vertebrae and placed in ice-cold DMEM/F12 medium. The tissues were cut into pieces < 1 mm in size and then treated with an enzyme solution containing 5 mg/ml dispase and 1 mg/ml collagenase at 37°C for 1 hr. After trituration and centrifugation, the cells were washed in 15% BSA, resuspended in DMEM/F12 containing 10% FBS, plated on glass coverslips coated with poly-D-lysine and laminin, cultured in an incubator at 37°C, and used within 24 hr of plating.

Rat DRG neurons were electroporated as follows. After washing the neurons with 15% BSA, the neurons were resuspended in DMEM/F12 and electroporated using a P3 Primary Cell 4D-Nucleofector X Kit L (cat. # V4XP-3012, Lonza) in accordance with the manufacturer's instructions. After electroporation, the neurons were cultured for 72 hr before use in order to allow the transgenes to express.

## Ca$^{2+}$ imaging

For Ca$^{2+}$ imaging experiments, cells were loaded at 37°C for 1 hr with 10 µg/ml Fluo-8 AM (AAT Bioquest, Inc) supplemented with 0.01% Pluronic F-127 (w/v; Invitrogen). Bile acids, bio-mimicked bile acid mixes, and/or various drugs to be tested were added to the cells in a chamber containing a custom-made 8-channel perfusion valve control system. Fluorescence images were acquired using a Nikon A1 confocal microscope.

## In situ hybridization and immunostaining of human, monkey, and mouse DRGs

All procedures working with postmortem human, monkey, and mouse tissues were previously approved according to the NIH guidelines and institute regulations. An animal protocol was approved by the University of Pennsylvania IACUC (institutional Animal Care and Use Committee) for the Luo lab to work with mice and mouse tissues. In addition, a standard operation procedure (SOP) was approved by the UPenn EHRS (Environmental Health and Radiation Safety) to guide the Luo lab to work with human and monkey tissues and wastes. We carefully followed the approved animal protocol and SOP.

Briefly, we purchased lumber DRGs (L4/L5) from healthy human donor from the National Disease Research Interchange (NDRI). Basically, a NDRI site will extract postmortem human donor DRGs and shipped them in cold culture medium with antibiotics to the Luo lab overnight. The monkey DRGs were extracted by Penn Veterinarians and stocked in cold culture medium with antibiotics (1X Penicillin-Streptomycin). The mouse DRGs were dissected from wild type mice (no known genetic modifications, mixed genetic background, mostly C57) in the Luo lab. The human, monkey, and mouse DRGs were frozen immediately in O.C.T. upon arrival, and frozen tissue sections (20 µm thickness) were prepared and stocked at −80 freezer. We performed single colormetric in situ hybridization, immunostaining, and RNAscope in situ hybridization with sections of multiple DRGs from two human donors, one monkey, and three mice.

Single colorimetric in situ hybridization in human, monkey, and mouse DRG sections was performed as follows. The fresh frozen sections were post fixed in freshly prepared 4% paraformaldehyde (PFA) in PBS for 20 min at RT, and then washed in fresh-DEPC PBS (1:1000 DEPC was added to 1x PBS immediately before use) and DEPC-pretreated PBS (1:1000 DEPC in PBS overnight, followed by autoclaving) for 10 min each. The sections were then immersed in a DEPC-containing antigen-retrieval solution containing 10 mM citric acid, 0.05% Tween-20 (pH 6.0) in a 95°C water bath for 20 min, and then cooled at RT for 30 min. After washing in DEPC-pretreated PBS for 10 min, the sections were incubated in a Proteinase K solution (25 µg/mL in DEPC-pretreated water) for 20 min and then washed in fresh-DEPC PBS and DEPC-pretreated PBS (10 min each). The sections were incubated in freshly prepared acetylation solution containing 0.1 M TEA and 0.25% acetic anhydride in DEPC-pretreated water for 10 min at RT, followed by a 10 min wash in DEPC-pretreated PBS. The prehybridization step was performed in probe-free hybridization buffer consisting of 50% formamide, 5x SSC, 0.3 mg/ml yeast tRNA, 100 µg/ml heparin, 1x Denhardt's solution, 0.1% Tween-20, 0.1% CHAPS, and 5 mM EDTA in RNase-free water at 62°C for 30 min in a humidified chamber, followed by an overnight hybridization step in hybridization buffer containing 5 ng/µl DIG-labeled riboprobes at 62°C in a humidified chamber (under a Parafilm coverslip). After the hybridization step, the sections were washed in 0.2x SSC at 68°C (once for 15 min and twice for 30 min each), followed by blocking in PBS containing 0.1% Triton X-100% and 20% horse serum for 1 hr at RT. The sections were then stained overnight at 4°C with pre-absorbed AP-conjugated sheep anti-DIG antibody (1:1000, Roche, cat. 11093274910) in PBS containing 0.1% Triton X-100% and 20% horse serum. The sections were washed 3 times for 10 min each in PBS containing 0.1% Triton X-100, followed by overnight incubation in the dark in AP buffer containing 100 mM Tris (pH 9.5), 50 mM MgCl2, 100 mM NaCl, 0.1% Tween-20, 5 mM levamisole, 0.34 mg/ml NBT (Roche cat. # 11383213001), and 0.17

mg/ml BCIP (Roche, cat. # 1138221001) to allow the color reaction to develop. The sections were washed 3 times for 10 min each in PBS, and then fixed for 30 min in 4% PFA in PBS. The sections were quickly rinsed 5 times in ddH2O, dried at 37°C for 1 hr, and dehydrated in xylene (3 times for 2 min each). Finally, the sections were mounted under a glass coverslip using Permount (Fisher).

Immunostaining was performed using a rabbit anti-hMRGPRX4 antibody (Abcam, cat. # ab120808), rabbit anti-hTGR5 antibody (Thermofisher, cat. # PA5-27076) and mouse anti-NeuN antibody (Sigma-Aldrich, cat. # MAB377). The fresh frozen human, monkey, or mouse sections were fixed in freshly prepared 4% PFA in PBS for 20 min at RT and then washed in PBS containing 0.1% Triton X-100 3 times for 10 min each, followed by block in PBS containing 0.1% Triton X-100% and 20% horse serum for 1 hr at RT. The sections were then incubated overnight in primary antibody at 4°C, washed with PBS containing 0.1% Triton X-100 3 times for 15 min each, and incubated with secondary antibody for 1 hr at RT. After washing with PBS 3 times for 15 min each, the sections were mounted under glass coverslips and Fluoromount-G (Invitrogen).

## RNAscope in situ hybridization

RNAscope in situ hybridization was performed in accordance with the manufacturer's instructions (Advanced Cell Diagnostics). In brief, fresh frozen human DRG sections were fixed, dehydrated, and treated with protease. The sections were then hybridized with the respective target probe for 2 hr at 40°C, followed by four-round signal amplification. The sections were then mounted under coverslips, sealed with nail polish, and stored in the dark at 4°C until imaged.

## Human itch test

The human itch test studies were approved by the Committee for Protecting Human and Animal Subjects at the Department of Psychology, Peking University (#2018-05-02). Volunteers were students and faculty members recruited from Peking University. All subjects provided written informed consent and were provided with the experimental protocol. All injections were performed using an INJEX 30 needle-free injection system (INJEX Pharma GmbH, Berlin, Germany). We performed two studies as described below.

In the first study (to measure bile acid–induced itch sensation), each tested compound was dissolved in physiological saline containing 7% Tween-80 (Sigma-Aldrich). The injection sites were cleaned with rubbing alcohol, and 25 µl of each solution was injected intradermally on the volar surface of each arm. The same volume of vehicle (saline containing 7% Tween-80) served as the negative control. Itch was defined as the desire to initiate scratching during the experiment, and the subjects rate the perceived intensity according the generalized labeled magnitude scale (LMS) described by *Green et al. (1996)*.

In the second study (to measure the effect of antihistamines on DCA-induced itch), two experimental sessions were performed, separated by 2 weeks, with 14 and 12 subjects participating in the first and second sessions, respectively. Approximately 1.5 g of topical antihistamine cream (doxepin hydrochloride cream, Chongqing Huapont Pharm. Co., China) or a placebo cream (cold cream, Eau Thermale Avène, Paris, France) was applied 2.5 hr before injection of DCA or histamine (Sigma-Aldrich); any unabsorbed cream was removed with alcohol. A 500 µg/25 µl solution of DCA was prepared as described above, and a 2.5 µg/25 µl solution of histamine was dissolved in saline; 25 µl of the DCA or histamine solution was injected into the volar surface of the arm as described above. In the first session, each subject received two intradermal injections of DCA (one at the antihistamine-treated site and one at the placebo-treated site). In the second session, each subject received two intradermal injections of histamine (one at the antihistamine-treated site and one at the placebo-treated site). The subjects then rate the itch sensation as described above.

## Quantification of plasma bile acids and bilirubin

These experiments were approved by the Committee for Biomedical Ethics, Peking University First Hospital (2017-R-94). Itch intensity was measured using a self-report numerical rating scale (NRS) (*Jenkins et al., 2009*), and whole blood samples were collected from patients with skin diseases and patients with liver diseases. Plasma was obtained by centrifuging 2 ml of whole blood at 4°C, 11,000 g for 10 min; 100 µl of each plasma sample was then mixed with 400 µl acetonitrile and left to sit at 4°C for 20 min. The mixture was centrifuged, and the supernatant was dried in a rotatory evaporator

(45°C under vacuum), and the dried residue was retrieved and dissolved in 60% methanol for further analysis.

The bile acid level in plasma samples was measured using HPLC-MS/MS (Agilent model LC1260 QQQ 6495). Chromatographic separation was performed in an ACQUITY UPLC HSS T3 column (2.1 mm × 100 mm, 1.8 μm; Waters Corp.). The mobile phase consisted of solution A (water) and solution B (acetonitrile). The total running time was 23 min, and a linear gradient (0.3 ml/min) was applied as follows: 0–2 min: 10% B - 40% B; 2–18 min: 40% B - 50% B; 18–19 min: 50–100% B; 19–20 min: 100% B; 20–21 min: 100–10% B; 21–23 min: 10% B. The injection volume was 5 μl, and the mobile phase flow rate was 3 ml/min. Deoxycholic-2,2,4,4,11,11-d6 acid (Sigma-Aldrich, cat. # 809675) was used as an internal standard.

Total bilirubin and direct bilirubin values were obtained from the patients' hospital blood chemistry reports.

## Statistical analysis

Summary data are presented as the mean ± s.e.m.. Human subjects were randomly assigned to control and experimental groups, and the subjects and investigators were double-blinded with respect to the experiment treatments. Data were analyzed using the Student's $t$-test, two-proportion z-test, Chi-square test, Fisher's exact test or One-way ANOVA and differences with a $P$-value of $< 0.05$ were considered significant.

## Acknowledgements

We thank Dr. Y Rao for sharing the tissue culture room, Dr. JH Zhao for collecting clinical blood samples. We are grateful to Dr. LQ Luo and Dr. Y Song for critical reading of the manuscript. We thank National Center for Protein Sciences at Peking University in Beijing, China, for assistance with the quantification of plasma bile acids and Dr. H Li for help with data analysis. We thank the National Disease Research Interchange (NDRI) for supply of normal adult human DRGs. We owe greatly to all human donors who donate their postmortem tissues for scientific research. We thank Erin Bote at the University of Pennsylvania and Dr. Norman Wiltshire at the Children Hospital of Philadelphia (CHOP) for helping with monkey DRG extraction. This work was supported by the Junior Thousand Talents Program of China and funding from Chinese Institute for Brain Research (Z181100001518004) to YL, and internal funding from the University of Pennsylvania supported work related to this project in WL's lab.

## Additional information

### Funding

| Funder | Grant reference number | Author |
|--------|------------------------|--------|
| Department of the Central Committee of the CPC | Junior Thousand Talents Program of China | Yulong Li |
| Chinese Institute for Brain Research | Z181100001518004 | Yulong Li |
| University of Pennsylvania | Internal funding | Wenqin Luo |

The funders had no role in study design, data collection and interpretation, or the decision to submit the work for publication.

### Author contributions

Huasheng Yu, Conceptualization, Resources, Data curation, Formal analysis, Supervision, Funding acquisition, Validation, Investigation, Visualization, Methodology, Writing—original draft, Project administration, Writing—review and editing; Tianjun Zhao, Conceptualization, Resources, Data curation, Formal analysis, Supervision, Validation, Investigation, Visualization, Methodology, Project administration, Writing—review and editing; Simin Liu, Conceptualization, Data curation, Formal analysis, Supervision, Validation, Investigation, Visualization, Methodology, Writing—original draft, Writing—review and editing; Qinxue Wu, Omar Johnson, Data curation, Formal analysis, Validation,

Investigation, Visualization, Methodology; Zhaofa Wu, Zihao Zhuang, Formal analysis, Validation, Investigation; Yaocheng Shi, Renxi He, Formal analysis, Investigation; Luxin Peng, Validation, Investigation; Yong Yang, Jianjun Sun, Resources, Investigation; Xiaoqun Wang, Haifeng Xu, Zheng Zeng, Resources; Peng Zou, Validation; Xiaoguang Lei, Resources, Writing—review and editing; Wenqin Luo, Conceptualization, Resources, Supervision, Project administration, Writing—review and editing; Yulong Li, Conceptualization, Resources, Supervision, Funding acquisition, Project administration, Writing—review and editing

### Author ORCIDs
Huasheng Yu https://orcid.org/0000-0001-5641-2512
Peng Zou http://orcid.org/0000-0002-9798-5242
Wenqin Luo https://orcid.org/0000-0002-2486-807X
Yulong Li https://orcid.org/0000-0002-9166-9919

### Ethics

Human subjects: All experiments involving human subjects have been approved by institutional review board or ethics committee and the informed consent, and consent to publish, was obtained. Collection of DRG tissue from adult humans was approved by the Committee for Medical Science Research Ethics, Peking University Third Hospital (IRB00006761-2015238), and collection from human embryos was approved by the Reproductive Study Ethics Committee of Peking University Third Hospital (2012SZ-013 and 2017SZ-043) and Beijing Anzhen Hospital (2014012x). The human itch test studies were approved by the Committee for Protecting Human and Animal Subjects at the Department of Psychology, Peking University (#2018-05-02). Collection of blood samples from patients were approved by the Committee for Biomedical Ethics, Peking University First Hospital (2017-R-94).

### Decision letter and Author response
Decision letter https://doi.org/10.7554/eLife.48431.025
Author response https://doi.org/10.7554/eLife.48431.026

## Additional files

### Supplementary files
• Supplementary file 1. Genes that are highly expressed in human DRG.
DOI: https://doi.org/10.7554/eLife.48431.020
• Supplementary file 2. GPCRs expression profiling in human DRG. Red labeled genes are candidate GPCRs that are highly expressed in human DRG. The blue labeled gene is TGR5.
DOI: https://doi.org/10.7554/eLife.48431.021
• Transparent reporting form
DOI: https://doi.org/10.7554/eLife.48431.022

### Data availability
All data generated or analysed during this study are included in the manuscript and supporting files.

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
