## [Decision Letter]

Thank you for submitting your article "MRGPRX4 is a novel bile acid receptor in cholestatic itch" for consideration by *eLife*. Your article has been reviewed by three peer reviewers, and the evaluation has been overseen by a Reviewing Editor and Gary Westbrook as the Senior Editor. The following individual involved in review of your submission has agreed to reveal their identity: Diana M Bautista (Reviewer #3). The reviewers have discussed the reviews with one another and the Reviewing Editor has drafted this decision to help you prepare a revised submission.

Summary:

In this work by Yu and colleagues, the authors examine the important question of why patients with liver disease often times develop cholestatic pruritus. Previous studies have shown that bile acids can trigger acute itch in mice via TGR5 signaling in a subset of mouse DRG neurons (Alemi et al., 2013) and that MRGPRX4 is activated by bilirubin (Meixiong et al., 2019) and bile acids (Meixiong et al, 2019). Here the authors probe the molecular mechanisms by which bile acids promote itch in human subjects and DRG neurons. To address the specific receptor for itch-inducing substances, they examined orphan G-protein coupled receptors that are highly expressed in human sensory afferents identified via public expression databases. Specifically, they identified 7 highly expressed GPCRs which they expressed in heterologous cells and assayed for sensitivity to bile extracts using G-protein coupled assays.

In this approach they focused on MRGPRX4 as a candidate receptor because of its lone sensitivity to bile acids in a Gq-dependent assay. Although this is a significant finding, as the authors point out, MRGPRX4 was recently shown by Meixiong et al. to be activated by bile acids and lead to itch when the human receptor was expressed in mice. Nonetheless, the authors expand upon these and other previous findings to fully define the role of Mrgprx4 in human tissues, as well as identify the specific ligands that may act on this receptor to induce itch. They show that bile acids are more robust activators of MRGPRX4, and itch in human subjects, compared to bilirubin, and that bilirubin potentiates bile acid activation of MRGPRX4. The inclusion of human and primate assays is a strength, as are their findings suggesting a difference in Mrgprx4 function and expression in human versus rodent dorsal root ganglia. Overall this is a comprehensive and well done study that will be of importance to the field.

Essential revisions:

1) In Figure 2B the authors focus on fractions 4 and 6 when clearly fractions 2, 3, and 5 have high activity. It is unclear why this was done. Can the authors examine the potential for itch inducing substances in these fractions, or at least confirm that the relevant compound DCA found in fraction 4 is also in these other active fractions?

2) In Figure 3K the expression of the P180A mutant is clearly 3-fold higher than the other mutants, a point that should be evaluated or discussed.

3) In addition to the RNAscope representative images in Figure 4F, some examples of the entire section should be displayed showing colocalization of MRGPRX4 with other genes as this is the main method used to define the MRGPRX4 neuronal population.

4) It's unclear how the representative Fluo-8 images in Figure 4H, left panel, are related to the small pseudocolor images under the traces Figure 4H, right panel. Full pseudocolor images of the DRGs with and without bile acids should be included (same as the format of Figure 4H, left panel).

5) In prior work, bilirubin was shown to not only trigger calcium influx via MRGPRX4 but also action potential firing in rodent neurons. Does bile acid activation of MRGPRX4 drive excitability of DRG neurons? Since the manuscript only reports relative fluo-8 fluorescence changes, the magnitudes of the bile-acid evoked calcium responses in neurons are unknown. Please discuss this point.

6) Do adult and embryonic DRG neurons display similar MRGPRX4 expression patterns and bile acid responses? Is the data currently pooled in the summary figures? If so, some statistical analyses should be performed to ensure there are no age-specific differences before combining the data.

7) The bilirubin modulation data should be included as a main figure in the manuscript.

8) What antibodies and criteria were used to define satellite glia cells in the IHC experiments? Please provide this information.

9) The title should be changed to avoid the word "novel" because MRGPRX4 has already been implicated as a receptor that mediates cholestatic itch.

---

## [Author Response]

Essential revisions:1) In Figure 2B the authors focus on fractions 4 and 6 when clearly fractions 2, 3, and 5 have high activity. It is unclear why this was done. Can the authors examine the potential for itch inducing substances in these fractions, or at least confirm that the relevant compound DCA found in fraction 4 is also in these other active fractions?

We thank the reviewer for raising this question. We have noticed that the fractions 2, 3 and 5 also have high activity. Using ^1^H-NMR, we found that fractions 2, 3 also contained the characteristic peaks of bile acids (Figure 2—figure supplement 1), suggesting that the activity may come from the bile acids in these fractions. Due to the limited amount of material, we haven’t been able to test fraction 5 using ^1^H-NMR, but we suspect the activity of this fraction may also come from bile acids. We modified text to explain this point in the revised manuscript.

2) In Figure 3K the expression of the P180A mutant is clearly 3-fold higher than the other mutants, a point that should be evaluated or discussed.

We thank the reviewer for noticing this finding. In this experiment, we generated stable cell lines expressing wild-type and mutant receptors via piggyBac transposon system. The gene is randomly inserted into the genome of the HEK293T cell lines. Thus, both insertion sites and copy numbers could significantly affect the expression level. The P180A mutant gene might be inserted into a favorable genomic site or multiple copies of the gene might be inserted into the genome, which leads to much higher membrane expression level than WT. Nevertheless, P180A mutant still shows less response to DCA. We hope reviewers could agree that P180 in MRGPRX4 is an important amino acid residue for bile acid sensing. We have added the discussion to the manuscript as the reviewer suggested.

3) In addition to the RNAscope representative images in Figure 4F, some examples of the entire section should be displayed showing colocalization of MRGPRX4 with other genes as this is the main method used to define the MRGPRX4 neuronal population.

We thank the reviewer for the good suggestion. There is a technical reason we didn’t include the low magnification pictures in the initial submission. MRGPRX4 signals in most positive DRG neurons are rather weak (have ~20 small fluorescent dots, indicated by arrow heads in Author response image 1 and as shown in the Figure 4F of the main text), and it is hard to tell a real positive neuron from background (big fluorescent dots positive for all three channels (indicated by arrows), which unfortunately exist for RNAscope in situ technique, especially for the weak probe channel) at the low magnification (Author response image 1 upper row). We thus mostly took high magnification images to be sure. Therefore, we have included a large-scale image of the DRG section, in which triple RNAscope in situ hybridization of *TRPV1, H1R1*, and *MRGPRX4* were performed (Author response image 1). As the reviewers can see, the TRPV1+ (usually more than 100 positive dots per cells) and H1R1+ (usually more than 50 to 100 positive dots per cells) neurons are easy to tell (indicated by arrowheads), but it is much harder to tell positive MRGRPX4+ neurons at this low magnification.

**Author response image 1. respfig1:** Representative large-scale RNAscope images of human DRG. Each fluorescent dot indicated by an arrowhead represents one single mRNA transcript. The large fluorescent dots indicated by arrows are non-specific background (positive for all three channels). Scale bar, 100 μm and 50 μm.

4) It's unclear how the representative Fluo-8 images in Figure 4H, left panel, are related to the small pseudocolor images under the traces Figure 4H, right panel. Full pseudocolor images of the DRGs with and without bile acids should be included (same as the format of Figure 4H, left panel).

We thank the reviewer for this good suggestion and we apologize for the unclear presentation. We have included a representative full pseudocolor image to show the responses of all imaged DRG neurons when applying DCA and other compounds (Figure 5G, Figure 5—figure supplement 2A). The circled cells on the left panel correspond to the representative traces and small pseudocolor images on the right panel.

5) In prior work, bilirubin was shown to not only trigger calcium influx via MRGPRX4 but also action potential firing in rodent neurons. Does bile acid activation of MRGPRX4 drive excitability of DRG neurons? Since the manuscript only reports relative fluo-8 fluorescence changes, the magnitudes of the bile-acid evoked calcium responses in neurons are unknown. Please discuss this point.

Since the expression percentage of MRGPRX4 is relatively low (6-7%) in human DRG neurons, and the quality of adult human DRG tissues from schwannoma patients, which were used for calcium imaging in this manuscript, are very limited, it is very difficult, if not impossible, to detect bile acid induced firing in the MRGPRX4-positive human DRG neurons. Alternatively, we electrically transfected MRGPRX4 in cultured rat DRG neurons and detected bile acid-induced action potentials of in MRGPRX4-expressing rat DRG neurons but not in non-transfected control cells by electrophysiology (Figure 5E, F, Figure 5—figure supplement 1E, F), demonstrating that activation of MRGPRX4 by bile acid could drive excitability of DRG neurons. In addition, we compared the activity of bilirubin and DCA to MRGPRX4 in cultured rat DRG neurons. Similar to the results in HEK293T cells. DCA showed stronger activity than bilirubin as it induced larger Ca^2+^ response both in amplitude and percentage in MRGPRX4-expressing neurons (Figure 9D-G). We have included these results in the revised manuscript.

6) Do adult and embryonic DRG neurons display similar MRGPRX4 expression patterns and bile acid responses? Is the data currently pooled in the summary figures? If so, some statistical analyses should be performed to ensure there are no age-specific differences before combining the data.

We thank the reviewer for raising this point and we apologize for the confusion. Given that the difficulties to obtain the human DRG tissues, we haven’t directly examined the MRGPRX4 expression patterns in the embryonic DRG tissues. Nevertheless, the percentage (5.2%,Figure 5H) of cultured embryonic DRG neurons responding to bile acids is close to expression percentage (6-7%) of MRGPRX4 in the adult DRG tissues. For cultured adult DRG cells, the percentage responding to bile acids varied from batch to batch (3.3-18.2%, Figure 5—figure supplement 2D1-3). The adult DRG tissues are derived from schwannoma patients undergoing surgical excision (as described in the Materials and methods), so the different tissue qualities may contribute to the huge variability. We agree that it is not appropriate to combine the adult and embryonic data together for statistical analysis. We have statistically analyzed each culture separately (Figure 5G, HFigure 5—figure supplement 2) and revised our text to clarify this point.

7) The bilirubin modulation data should be included as a main figure in the manuscript.

Thank you for digging out and highlighting the bilirubin modulation data. We have included this part as a main figure (Figure 9).

8) What antibodies and criteria were used to define satellite glia cells in the IHC experiments? Please provide this information.

We mainly used morphological criteria to define human and monkey satellite glia cells. We tried immunostaining of a few antibodies (TLR4 for human and monkey satellite glia cells, HTA125, eBioscience; GPAP for mouse satellite glia cells, GFAP, AB_2313547 Aves Labs), which may stain satellite glia cells based on literatures^1,2^, but none of them work for the human and monkey DRG satellite glia cells at our hand. This could be due to different tissue fixation and processing condition. We used fresh frozen for human DRGs, which works nicely for RNAscope in situ hybridization and antibody staining of MRGPRX4 and TGR5. Instead, we co-labeled human DRG neurons with NeuN antibody. We found that TGR5 immunostaining signal does not overlap with NeuN but surrounding these neurons and that TGR5 in situ signals also surround big negative DRG neuron soma. Given the size and morphology of satellite glia cells published before^3^, we believe that the surrounded cells are satellite glia cells. Of course, if we had a good antibody to specifically label human/monkey satellite glia cells, we would be able to provide even stronger evidence.

9) The title should be changed to avoid the word "novel" because MRGPRX4 has already been implicated as a receptor that mediates cholestatic itch.

We agree that we should avoid the word “novel” in title since MRGPRX4 has already been implicated as a receptor that mediates cholestatic itch. We change our title to “MRGPRX4 is a bile acid receptor for human cholestatic itch”.

References:

1) Mitterreiter, J. G. et al. Satellite glial cells in human trigeminal ganglia have a broad expression of functional Toll-like receptors. Eur J Immunol 47, 1181-1187, doi:10.1002/eji.201746989 (2017).

2) Prabhakar, A., Vujovic, D., Cui, L., Olson, W. & Luo, W. Leaky expression of channelrhodopsin-2 (ChR2) in Ai32 mouse lines. PLoS One 14, e0213326, doi:10.1371/journal.pone.0213326 (2019).

3) OpenStax Anatomy and Physiology. Version 8.25 edn, (2016).